# Training Data Efficiency in Multimodal Process Reward Models

Jinyuan Li[1]  Chengsong Huang[1]  Langlin Huang[1]  Shaoyang Xu[2]  Haolin Liu[3]  Wenxuan Zhang[2]
Jiaxin Huang[1]

## Abstract

Multimodal Process Reward Models (MPRMs) are central to step-level supervision for visual reasoning in MLLMs. Training MPRMs typically requires large-scale Monte Carlo (MC)-annotated corpora, incurring substantial training cost. This paper studies the data efficiency for MPRM training. Our preliminary experiments reveal that MPRM training quickly saturates under random subsampling of the training data, indicating substantial redundancy within existing MC-annotated corpora. To explain this, we formalize a theoretical framework and reveal that informative gradient updates depend on two factors: label mixtures of positive/negative steps and label reliability (average MC scores of positive steps). Guided by these insights, we propose the *Balanced-Information Score* (BIS), which prioritizes both mixture and reliability based on existing MC signals at the rollout level, without incurring any additional cost. Across two backbones (InternVL2.5-8B and Qwen2.5-VL-7B) on VisualProcessBench, BIS-selected subsets consistently match and even surpass the full-data performance at small fractions. Notably, the BIS subset reaches full-data performance using only 10% of the training data, improving over random subsampling by a relative 4.1%. Our code is released Balanced-Info-MPRM.

## 1. Introduction

Process Reward Models (PRMs) (Ma et al., 2023; Zhu et al., 2025; Tan et al., 2025) provide step-level supervision for reasoning by scoring intermediate steps instead of only the final answer. In multimodal reasoning, Multimodal PRMs

[1]Washington University in St. Louis [2]Singapore University of Technology and Design [3]University of Virginia. Correspondence to: Jinyuan Li <ljinyuan@wustl.edu>, Jiaxin Huang (Corresponding Author) <jiaxinh@wustl.edu>.

*Proceedings of the 43rd International Conference on Machine Learning*, Seoul, South Korea. PMLR 306, 2026. Copyright 2026 by the author(s).

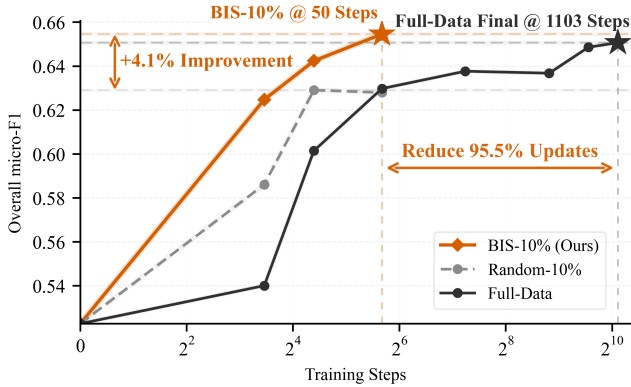

*Figure 1.* Overall micro-F1 on VisualProcessBench using InternVL2.5-8B trained via full training set or different subsets. Our BIS-10% effectively matches the final performance of Full-Data setting using $10\times$ fewer rollouts.

(MPRMs) are increasingly used for Multimodal Large Language Models (MLLMs) (Wang et al., 2024e;f; Bai et al., 2023; 2025; Liu et al., 2023; Team et al., 2024; 2025) to conduct complex visual reasoning tasks both during training and at test time (Wang et al., 2025a;b; Zhang et al., 2025a; Luo et al., 2024; Du et al., 2025; Tu et al., 2025; Cao et al., 2025; Dong et al., 2025). Common practices for training MPRMs generally rely on large-scale Monte Carlo (MC)-annotated rollouts (e.g., VisualPRM400K-v1.1 (Wang et al., 2025b), with 565K rollouts and 3.17M annotated steps), which makes training computationally expensive. In this paper, we study the practical bottleneck in **training data efficiency** for MPRMs: how does MPRM performance scale with the rollout budget, and how can we select informative subsets that preserve full-data performance?

Our preliminary study suggests substantial redundancy in MC-annotated MPRM training data. We randomly subsample the training data at varying fractions $\rho$ and find that performance quickly saturates at small $\rho$, with a moderate gap to the full dataset. This trend persists even when the subset is trained longer to match full-data training steps. We further compare several size-matched heuristic subsets and find that selecting rollouts that mix correct and incorrect steps is more informative than random selection, whereas rollouts with the lowest average MC scores tend to contain noisy pseudo-positive labels and hurt performance. This suggests two key

criteria for high-quality rollouts: **mixture** and **reliability**.

To substantiate this intuition, we formalize a teacher–student abstraction framework for theoretical analysis, and connect the interplay between gradient signal, label noise and data redundancy. We model MC estimation noise via a probabilistic label-flip model and show how it affects training gradients. This modeling supports the view that MPRM training is primarily limited by gradient noise rather than data scarcity. Moreover, our theory explains why mixture and reliability capture rollout quality: **mixture** tracks model uncertainty, while **reliability**, measured by MC scores, captures the noise level in positive steps. Their contributions interact multiplicatively in shaping informative gradients.

Building on these insights, we introduce the *Balanced-Information Score* (BIS), a rollout-level criterion that instantiates the "mixed but reliable" principle. BIS quantifies both label mixture (of positive and negative steps) and reliability (average MC score over positive steps). It is model-agnostic and only uses the MC signals stored in the dataset, without requiring extra model calls. Extensive experiments with two backbones (InternVL2.5-8B (Chen et al., 2024c) and Qwen2.5-VL-7B (Bai et al., 2025)) on VisualProcess-Bench (Wang et al., 2025b) show that BIS recovers full-data performance at small subset ratios, with the largest gain over random sub-sampling in low-budget regimes. In particular, Figure 1 shows that a BIS-selected 10% subset trained for only 50 steps suffices to reach and even surpass the full-data performance on InternVL2.5-8B, saving 95.5% computational cost. Taken together, these findings provide a practical recipe with grounded analysis for reducing training compute for MPRMs without sacrificing model performance.

## 2. Preliminary Study

### 2.1. Background and General Setup

Previous MPRM research mainly improves supervision pipelines or training frameworks (detailed in Appendix A). In contrast, we study post-hoc rollout selection with no extra supervision or compute. We adopt the standard MPRM training setup and keep it fixed throughout. Following prior works (Wang et al., 2024d; Zhang et al., 2025g), we use the VisualPRM400K-v1.1 dataset (Wang et al., 2025b), where each reasoning step is annotated with an MC-estimated success rate from $N{=}16$ sampled continuations. Step labels are binarized, so $y_t{=}1$ if the MC score $> 0$ (at least one continuation reaches correct final answer), and $y_t{=}0$ otherwise. Specifically, for a reasoning rollout with $T$ steps, a special token `<prm>` is appended after each step $t$. The model is trained to predict the step-level probability ("Yes"/"No") tokens using the cross-entropy loss $\mathcal{L} = -\sum_{t=1}^{T}(y_t \log p_t + (1-y_t)\log(1-p_t))$, where $p_t$ is the probability of predicting the token "Yes". We use

*Table 1.* Dataset statistics for different training-set settings. "Steps" denote reasoning steps with annotated labels.

| Metric | Full-Data | Random-25% | Low-MC-25% | Mixed-25% |
|---|---|---|---|---|
| # rollouts | 565,096 | 141,288 | 141,210 | 141,253 |
| # reasoning steps | 3,174,394 | 794,756 | 796,940 | 795,752 |
| Avg. steps/rollout | 5.62 | 5.63 | 5.64 | 5.63 |
| Avg. words/step | 27.8 | 27.8 | 29.9 | 27.6 |
| Error-step ratio | 3.57% | 3.61% | 12.57% | 11.02% |
| Avg. MC/step | 0.8566 | 0.8590 | 0.6010 | 0.7160 |

InternVL2.5-8B (Chen et al., 2024c) in preliminary study, and evaluate the MPRM performance on VisualProcess-Bench (Wang et al., 2025b), a human-annotated step-level benchmark spanning five sources (MathVision (Wang et al., 2024c), MathVerse (Zhang et al., 2024), MMMU (Yue et al., 2024), DynaMath (Zou et al., 2025), and WeMath (Qiao et al., 2025)), and follow its protocol to report per-source macro-F1 and micro-averaged F1 over all sources. Training details, data statistics are provided in Appendix C, B and H.

### 2.2. Random Sub-Sampling: Evidence of Redundancy

> 💡 **Empirical Finding 1:** MPRM performance quickly saturates under random subsampling, indicating strong redundancy in the training data.

To assess how MPRM performance scales with process-supervision data, we use the full training corpus as *Full-Data* and evaluate random subsampling. For any keep ratio $\rho$, *Random-$\rho$* retains a fraction $\rho$ of rollouts from each of the 38 source subsets, preserving their relative composition.

We train with single-pass fine-tuning and report micro-F1 for different Random-$\rho$ subsets in Figure 2a. Performance improves with $\rho$ but quickly plateaus, exhibiting pronounced diminishing returns. This suggests substantial redundancy in the MC-annotated rollouts, as discarding a large fraction of rollouts only modestly degrades performance.

To probe the plateau at moderate $\rho$, we take $\rho = 25\%$ and compare Random-25% with Full-Data under a matched compute budget. We match the number of training steps by training Random-25% for four epochs, making its training cost comparable to one epoch of Full-Data. Table 1 summarizes the corpus statistics for these settings.

We compare their learning curves in Figure 2b. Although the Full-Data model eventually performs better, the gap to Random-25% remains moderate, confirming substantial redundancy in the training data. We also report per-source results in Appendix D. Additionally, under matched updates, Random-25% slightly overfits and its performance degrades late in training. In the remaining experiments, we use single-pass fine-tuning, where each rollout is seen exactly once.

Given the redundancy above, a natural next question is: *is there a principled data selection method that substantially filters training data while preserving full-data performance?*

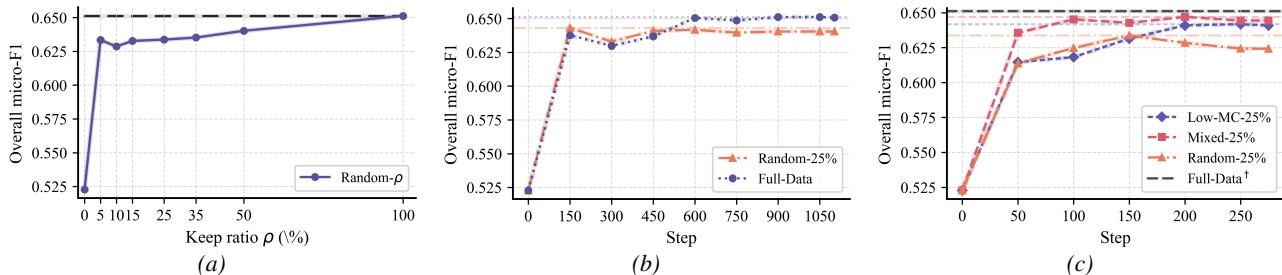

*Figure 2.* Overall VisualProcessBench micro-F1 under different data regimes. (a) Single-pass scaling with random sub-sampling; per-source macro-F1 curves are shown in Figure 4. (b) Training on Full-Data vs. Random-25% under matched updates; per-source macro-F1 curves are shown in Figure 5. (c) Training on three 25% subsets for one epoch; per-source macro-F1 curves are shown in Figure 6. Full-Data[†] denotes the best checkpoint of a one-epoch Full-Data run (4× more optimization steps than 25% subsets).

### 2.3. Characterizing Informative Rollouts

> 💡 **Empirical Finding 2:** Effective supervision comes from *mixed* rollouts that contain both correct and incorrect steps while maintaining *reliable* positive labels.

We now shift focus from *how many* rollouts to use to *which* rollouts to keep. To study the impact of increased exposure to negative steps, we construct three subsets of Visual-PRM400K: *Random-25%*, *Low-MC-25%*, and *Mixed-25%*.

**Random-25%** randomly samples 25% of rollouts from each source to preserve the original dataset distribution.

**Low-MC-25%** is constructed by ranking rollouts within each source by their average MC score per step and retaining the bottom 25%. As a result, the average MC per step drops to 0.601 and the incorrect-step ratio rises to 12.57%, far higher than in Random-25% (3.61%). Many low-MC steps have only a few successful continuations out of $N = 16$, yet are still labeled as positive under the standard binarization rule, making them prone to pseudo-positive labels.

**Mixed-25%** prioritizes rollouts with both positive and negative steps. Since mixed rollouts are only 7.67%, when a source has fewer than 25% mixed rollouts, we fill the remainder by randomly sampling from the rest. It has a similar incorrect-step ratio to Low-MC-25% (11.02% vs. 12.57%) but a higher average MC score (0.716 vs. 0.601), exposing the model to many negative steps while still anchoring them with reasonable amount of reliable positive labels.

Table 1 summarizes the statistics for these subsets. Using the same training protocol, we fine-tune each 25% subset for one epoch and plot the overall VisualProcessBench micro-F1 over training steps in Figure 2c. From this comparison, we observe two patterns as follows:

**First,** at their best checkpoints, the three subsets satisfy Mixed-25% > Low-MC-25% > Random-25%. Both Low-MC-25% and Mixed-25% outperform Random-25%, indicating that, under a fixed data budget, exposing the model to more incorrect steps is beneficial.

**Second,** Mixed-25% consistently yields the strongest performance even though its incorrect-step ratio is comparable to Low-MC-25% while its average MC score is notably higher. This suggests that neither maximizing negative steps nor minimizing average MC scores alone is sufficient. Extremely low-MC steps tend to be noisy pseudo-positives (labeled positive despite very low success rates), whereas rollouts that combine reasonably reliable positive steps with clear errors provide more useful supervision.

These observations motivate us to find a rollout-scoring mechanism that prioritizes two aspects: (1) emphasizing mixed rollouts (containing both correct and incorrect steps) while (2) avoiding noisy rollouts that contain many extremely low MC-score steps.

## 3. Theoretical Analysis

Before introducing our scoring mechanism, we provide a theoretical analysis to explain empirical findings 1 and 2. We formalize the interplay among data redundancy, label noise, and gradient behavior, which also guides the design of an effective data-selection score.

### 3.1. Teacher–Student Abstraction

We model MPRM training using a linear teacher–student framework: the teacher represents the ideal model that knows true step-level correctness, while the student model learns from noisy MC-annotated labels. We model step-level label prediction as logistic regression on the representation space $\phi$ for simplicity. For the $j$-th step in rollout $x$, let $\phi_{x,j} \in \mathbb{R}^d$ denote the hidden representation at the `<prm>` token position and $Y_{x,j}^{\text{true}} \in \{0, 1\}$ its binary label.

An ideal "teacher" MPRM is

$$q^*(\phi) = \Pr(Y^{\text{true}} = 1 \mid \phi) = \sigma(\langle w^*, \phi \rangle), \quad (1)$$

where $w^* \in \mathbb{R}^d$ is the optimal parameter and $\sigma$ is the sigmoid function. The student MPRM (our learned model) is

$$q_w(\phi) = \sigma(\langle w, \phi \rangle), \quad (2)$$

trained by minimizing the expected logistic loss

$$\mathcal{L}(w) = \mathbb{E}_{(\phi,Y)}[-Y \log q_w(\phi) - (1-Y)\log(1-q_w(\phi))]. \quad (3)$$

In the MC-annotated training set, each step is associated with an MC score $s_{x,j} \in [0,1]$ from $N$ sampled continuations and a binary label $Y_{x,j}^{\mathrm{mc}} = \mathbb{I}[s_{x,j} > 0]$. For the theoretical analysis, we do not model the MC sampling explicitly; instead, we model $(\phi_{x,j}, Y_{x,j}^{\mathrm{true}})$ as i.i.d. samples from the teacher model in Eq. (1). The observed MC score $s_{x,j}$ (and the corresponding binarized training label $Y_{x,j}^{\mathrm{mc}}$) provides a noisy estimate of the underlying correctness probability $q^*(\phi_{x,j})$. Under this formulation, the student $q_w$ is trained with the logistic loss in Eq. (3) on the observed MC labels $Y_{x,j}^{\mathrm{mc}}$, matching the objective in Section 2.1.

## 3.2. Understanding the Plateau of Random Subsets

> 💡 **Theoretical Finding 1:** MPRM training mostly suffers from noisy gradients instead of insufficient training data.

In this part, we aim to explain why randomly sub-sampled subsets across varying $\rho$ recover much of the full-dataset performance. In the teacher–student setup of Section 3.1, we consider training the student model to minimize a logistic loss $\mathcal{L}(w)$, with $w^*$ being the optimal parameter (achieved with infinite data and infinite training), and $w_T$ the parameters after $T$ finite stochastic gradient descent (SGD) steps.

Under the assumptions in Appendix F.1, standard non-asymptotic analyses of SGD for logistic regression (Bach & Moulines, 2013; Bottou & Bousquet, 2007) yield a bound on the excess risk (the gap in expected loss between $w_T$ and $w^*$) of the form

$$\mathbb{E}[\mathcal{L}(w_T)] - \mathcal{L}(w^*) \lesssim C_{\mathrm{data}} N_{\mathrm{eff}}^{-1/2} + C_{\mathrm{opt}} T^{-1/2} \quad (4)$$

This bound contains two components: (1) a *data complexity term* $C_{\mathrm{data}} N_{\mathrm{eff}}^{-1/2}$, which decays with the effective sample size $N_{\mathrm{eff}}$, and (2) an *optimization error term* $C_{\mathrm{opt}} T^{-1/2}$, which decays with the number $T$ of SGD updates. $C_{\mathrm{data}}, C_{\mathrm{opt}} > 0$ are problem-dependent constants that do not scale with $N_{\mathrm{eff}}$ or $T$. A detailed derivation of Eq. (4) is given in Appendix F.1.

**Why larger datasets help less than expected.** In MC-annotated training data, many steps receive noisy labels, especially when only a few out of $N$ continuations succeed. This label noise increases the stochastic-gradient noise level, which effectively enlarges the constant $C_{\mathrm{opt}}$ in the optimization term (Moulines & Bach, 2011). Meanwhile, for VisualPRM400K-v1.1 with 3.17M annotated steps, the data complexity term $C_{\mathrm{data}} N_{\mathrm{eff}}^{-1/2}$ is relatively small. Taken together, the optimization term dominates the total error, and further increasing $N_{\mathrm{eff}}$ yields only marginal gains.

Now consider a random subset Random-$\gamma$ that keeps each data point with probability $\gamma \in (0,1)$, so the effective sample size becomes $\gamma N_{\mathrm{eff}}$ while the problem-dependent constants remain comparable. Let $T_\gamma$ denote the number of SGD updates. In the *matched-update* setting we keep the update budget fixed, $T_\gamma = T$, and Eq. (4) gives

$$\mathbb{E}[\mathcal{L}(w_{T_\gamma})] - \mathcal{L}(w^*) \lesssim \gamma^{-\frac{1}{2}} C_{\mathrm{data}} N_{\mathrm{eff}}^{-1/2} + C_{\mathrm{opt}} T^{-\frac{1}{2}}.$$

Random sub-sampling therefore amplifies only the (already small) data term by $\gamma^{-1/2}$, while leaving the optimization term unchanged; once $C_{\mathrm{data}} N_{\mathrm{eff}}^{-1/2} \ll C_{\mathrm{opt}} T^{-1/2}$, changing $\gamma$ has only a modest effect on the total error, explaining why Random-25% closely tracks Full-Data in Figure 2b.

In the *single-pass* setting we have $T_\gamma = \gamma T$, so Eq. (4) gives

$$\mathbb{E}[\mathcal{L}(w_{T_\gamma})] - \mathcal{L}(w^*) \lesssim C_{\mathrm{data}}(\gamma N_{\mathrm{eff}})^{-\frac{1}{2}} + C_{\mathrm{opt}}(\gamma T)^{-\frac{1}{2}}.$$

Let $B := C_{\mathrm{data}} N_{\mathrm{eff}}^{-\frac{1}{2}} + C_{\mathrm{opt}} T^{-\frac{1}{2}}$. The right-hand side can be written as $B_\gamma := \gamma^{-\frac{1}{2}} B$, so both terms are scaled by the $\gamma^{-1/2}$. For the full-data configuration with $\sim 3.17$M annotated steps and $\sim 1.1$k updates, we operate in a low-error regime where $B \ll \epsilon_{\mathrm{tar}}$. Here $\epsilon_{\mathrm{tar}}$ is the target error level. When $B$ is already far below $\epsilon_{\mathrm{tar}}$, multiplying it by the constant factor $\gamma^{-1/2}$ still gives $B_\gamma \leq \epsilon_{\mathrm{tar}}$, so both Full-Data and Random-$\gamma$ single-pass training remain within the desired accuracy range. In such a regime, a constant factor $\gamma^{-1/2}$ in the bound is not enough to induce a large performance gap, which matches the small empirical difference we observe in Figure 2c and reinforces that improving gradient quality, rather than merely enlarging $N_{\mathrm{eff}}$, is key to escaping the current optimization floor.

## 3.3. Why Mixed but Reliable Rollouts Are Informative?

Here, we explain why *mixture* and *reliability* characterize informative rollouts: label mixture tracks teacher uncertainty, while MC scores quantify the reliability of step labels.

### Step-level Information from Teacher Uncertainty

> 💡 **Theoretical Finding 2:** Ideal teacher-model uncertainty $q^*(\phi)(1 - q^*(\phi))$ *quantifies* per-step information.

Under the teacher–student framework in Section 3.1, the gradient of logistic loss for a step with representation $\phi \in \mathbb{R}^d$ and label $Y \in \{0,1\}$ is

$$g(\phi, Y; w) = (q_w(\phi) - Y)\phi, \qquad q_w(\phi) = \sigma(\langle w, \phi \rangle).$$

Since we study offline data selection, which is fixed throughout training, we measure how informative a step is under the teacher distribution rather than an evolving student, yielding a student-independent criterion. At the teacher's optimal parameter $w^*$, the second moment of the gradient has the form (derivation in Appendix F.2)

$$\mathbb{E}[\|g(\phi, Y; w^*)\|^2 \mid \phi] = q^*(\phi)(1 - q^*(\phi))\|\phi\|^2, \quad (5)$$

where $q^*(\phi) = q_{w^*}(\phi)$. $\mathbb{E}[\|g(\phi, Y; w^*)\|^2 \mid \phi]$ quantifies the expected per-step learning signal. Thus, for step-level

MPRM training, the most informative steps are those where the teacher is most uncertain ($q^*(\phi) \approx 1/2$), when the term $q^*(\phi)\big(1 - q^*(\phi)\big)$ reaches its maximum.

### Effect of Label Noise at Step Level

> 💡 **Theoretical Finding 3:** Extremely low-MC positive steps behave like label-reversed samples and produce gradients that carry little true signal and harm training.

To analyze how MC label noise affects learning, we adopt a symmetric label noise approximation: the true label $Y$ is flipped independently with probability $\eta \in [0, 1/2)$ to produce a noisy label $\tilde{Y}$, where restricting to $\eta < 1/2$ is without loss of generality since larger rates can be reduced to $1 - \eta$ by flipping the label semantics. In practice the effective noise is step-dependent, and we adopt this constant noise distribution to derive the explicit formula for the second moment of the noisy gradient at $w^*$, where $\tilde{g}(\phi, \tilde{Y}; w^*) = (q^*(\phi) - \tilde{Y})\phi$. A direct computation (Appendix F.3) yields

$$\mathbb{E}\big[\|\tilde{g}(\phi, \tilde{Y}; w^*)\|^2 \mid \phi\big]$$
$$= \Big((1 - 4\eta)\, q^*(\phi)\big(1 - q^*(\phi)\big) + \eta\Big) \|\phi\|^2. \quad (6)$$

Relative to the clean case in Eq. (5), the uncertainty term $q^*(\phi)\big(1 - q^*(\phi)\big)$ is shrunk by $(1 - 4\eta)$, with an additional $q^*$-independent noise $\eta\|\phi\|^2$. When $\eta$ is large, gradients are increasingly noise-dominated and carry little useful signal. Empirically, steps with extremely low-MC scores but positive labels are typically unstable: only a few out of all continuations succeed for incidental reasons (e.g., later self-correction), so these steps act like label-flipped negative steps rather than real positive steps.

### Rollout Label Mixture Estimates Teacher Uncertainty

> 💡 **Theoretical Finding 4:** Rollout label mixture $\hat{p}_x(1 - \hat{p}_x)$ is an $\mathcal{O}(1/n)$-biased estimator of the unobserved teacher-level uncertainty $\theta_x = \bar{q}_x(1 - \bar{q}_x)$ under noise-free labels.

We now relate rollout label mixture to the teacher's positive-label probabilities as follows. For a rollout $x$ with $n$ steps we define average step-wise information:

$$A(x) := \frac{1}{n} \sum_{j=1}^{n} q_{x,j}\big(1 - q_{x,j}\big).$$

Under the bounded-norm assumption in Appendix F.4, the unweighted and norm-weighted quantities differ by at most global multiplicative constants. For exposition, we begin with the teacher-consistent idealization $Y_{x,j} \mid q_{x,j} \sim$ Bernoulli($q_{x,j}$), under which $\{Y_{x,j}\}_{j=1}^{n}$ are unbiased samples from $\{q_{x,j}\}$. Let $\hat{p}_x := \frac{1}{n} \sum_{j=1}^{n} Y_{x,j}$ be the empirical positive-label fraction, and let $\bar{q}_x := \frac{1}{n} \sum_{j=1}^{n} q_{x,j}$ be the step-average teacher probability. Then $\hat{p}_x(1 - \hat{p}_x)$ is maximized when labels are balanced; it becomes 0 when the steps are all-positive or all-negative, so it directly measures label mixture within rollout $x$. By Lemma 2 in Appendix F.4,

conditioning on $\{q_{x,j}\}$ yields

$$\mathbb{E}[\hat{p}_x(1 - \hat{p}_x) \mid \{q_{x,j}\}] = \bar{q}_x(1 - \bar{q}_x) - \frac{1}{n^2} \sum_{j=1}^{n} q_{x,j}(1 - q_{x,j}). \quad (7)$$

By Jensen's inequality for $t \mapsto t(1 - t)$, we have

$$A(x) = \frac{1}{n} \sum_{j=1}^{n} q_{x,j}\big(1 - q_{x,j}\big) \leq \bar{q}_x(1 - \bar{q}_x) =: \theta_x. \quad (8)$$

Combining (7) and (8) yields the sandwich bound

$$\theta_x\Big(1 - \frac{1}{n}\Big) \leq \mathbb{E}\big[\hat{p}_x(1 - \hat{p}_x) \mid \{q_{x,j}\}\big] \leq \theta_x. \quad (9)$$

Thus $\hat{p}_x(1 - \hat{p}_x)$ is an observable estimate for the teacher-level uncertainty $\theta_x$, with only $\mathcal{O}(1/n)$ bias, and $A(x) \leq \theta_x$ by construction. Consequently, in this noise-free setting, rollouts that are nearly all-positive or all-negative have small $\hat{p}_x(1 - \hat{p}_x)$, suggesting smaller $\theta_x$ (and hence smaller expected $A(x)$). In contrast, rollouts with mixed labels tend to have larger $\theta_x$, which allows $A(x)$ to be larger and can yield gradient updates with stronger learning signal.

Now consider the symmetric flip noise model: $\tilde{Y}_{x,j}$ is obtained by flipping $Y_{x,j}$ with a constant rate $\eta$. Then $\tilde{Y}_{x,j} \mid q_{x,j} \sim$ Bernoulli($\tilde{q}_{x,j}$) with $\tilde{q}_{x,j} = (1 - 2\eta)q_{x,j} + \eta$ (Appendix F.4 Eq. (22)). Averaging over steps gives $\bar{\tilde{q}}_x = (1 - 2\eta)\bar{q}_x + \eta$, and the induced rollout-level mixture satisfies

$$\tilde{\theta}_x := \bar{\tilde{q}}_x(1 - \bar{\tilde{q}}_x) = (1 - 2\eta)^2\, \theta_x + \eta(1 - \eta). \quad (10)$$

Eq. (10) decomposes $\tilde{\theta}_x$ into a scaled uncertainty term $(1 - 2\eta)^2\theta_x$ plus an offset $\eta(1 - \eta)$. Since $\tilde{\theta}_x \approx \theta_x$ only for small $\eta$ (Appendix F.4 Eq. (24)), we next complement mixture with a reliability signal to identify low-noise rollouts.

### MC Scores as Effective Noise Indicators

> 💡 **Theoretical Finding 5:** MC scores monotonically reflect label reliability: low-MC positives exhibit high effective noise.

Mixture alone is not sufficient because positive labels can be noisy. We now model the MC annotation process and connect MC scores to the label-noise level. For step $j$ in rollout $x$, let $r_{x,j} \in [0, 1]$ be the probability that a single continuation from this step reaches the correct final answer, given its representation $\phi_{x,j}$. The MC annotator generates $N$ independent continuations, records the number of successful ones $K_{x,j}$, and stores the score $s_{x,j} = K_{x,j}/N$. Under the Binomial model $K_{x,j} \mid r_{x,j} \sim$ Binomial($N, r_{x,j}$) we have

$$\mathbb{E}[s_{x,j} \mid r_{x,j}] = r_{x,j}, \quad \mathrm{Var}(s_{x,j} \mid r_{x,j}) = \frac{1}{N}r_{x,j}(1 - r_{x,j}),$$

so $s_{x,j}$ is an unbiased estimator of $r_{x,j}$ and concentrates around it as $N$ grows. Under the standard binarization rule $Y_{x,j} = \mathbb{I}[K_{x,j} > 0]$, the resulting step-level probability of the positive label is

$$\Pr(Y_{x,j} = 1 \mid \phi_{x,j}) = 1 - (1 - r_{x,j})^N$$

which is strictly increasing in $r_{x,j}$. Since $r_{x,j}$ is deter-

mined by $\phi_{x,j}$, we model this probability using the teacher $q^*(\phi_{x,j}) = \sigma(\langle w^*, \phi_{x,j}\rangle)$ in Eq. (1). These observations link the MC score $s_{x,j}$, the binarized label $Y_{x,j}$, and the teacher probability $q^*(\phi_{x,j})$ through the underlying success probability $r_{x,j}$. We next quantify how this link translates into an effective noise level for positive labels.

To formalize reliability, we fix a threshold $\tau \in (0,1)$ and define a step to be $\tau$-*reliable* if its one-shot success probability is at least $\tau$, i.e., $Z_{x,j} := \mathbb{I}[r_{x,j} \geq \tau]$. We then relate this reliability notion to what the MC annotator actually observes. Specifically, we let the unobserved success probability $r_{x,j}$ vary across steps and model it with a Beta distribution $r_{x,j} \sim \text{Beta}(a,b)$. Conditional on $r_{x,j}$, the number of successful continuations among the $N$ MC samples satisfies $K_{x,j} \mid r_{x,j} \sim \text{Binomial}(N, r_{x,j})$. Under this Beta–Binomial model, observing $K_{x,j} = k$ yields the posterior $r_{x,j} \mid K_{x,j} = k \sim \text{Beta}(a+k, b+N-k)$. This induces an effective noise level for positive steps, defined as

$$\eta_{\text{eff}}(k) := \Pr(Z_{x,j} = 0 \mid K_{x,j} = k)$$
$$= \Pr(r_{x,j} < \tau \mid K_{x,j} = k) = I_\tau(a+k, b+N-k)$$

where $I_\tau(\cdot, \cdot)$ denotes the regularized incomplete beta function. For $K_{x,j} > 0$, $\eta_{\text{eff}}(k)$ is exactly the posterior probability that a positive step with $K_{x,j} = k$ is pseudo-positive, i.e., $\tau$-unreliable. Moreover, $\eta_{\text{eff}}(k)$ is strictly decreasing in $k$ (Lemma 3 in Appendix F.5). Consequently, low-MC positives ($K_{x,j} > 0$ but small $s_{x,j} = K_{x,j}/N$) have large $\eta_{\text{eff}}$ and are likely to be $\tau$-unreliable, so they behave like high-noise samples. Under the label-flipping noise model of Eq. (6), this corresponds to operating at a larger noise rate $\eta \approx \eta_{\text{eff}}(K_{x,j})$, which increases the noise term $\eta\|\phi_{x,j}\|^2$ and decreases the signal term $q^*(\phi_{x,j})(1 - q^*(\phi_{x,j}))\|\phi_{x,j}\|^2$.

Averaging $s_{x,j}$ over positive steps in a rollout thus yields a natural rollout-level reliability estimate: since $\eta_{\text{eff}}(k)$ decreases monotonically with the MC score $s_{x,j} = K_{x,j}/N$, rollouts whose positive steps have higher average MC scores also have smaller average $\eta_{\text{eff}}(K_{x,j})$, so their gradient updates are less affected by label noise.

**Mixture and Reliability Couple Multiplicatively**

> 💡 **Theoretical Finding 6:** Rollouts are most informative when *both* label mixture and reliability are high.

Finally, we combine the previous two parts and analyze how label mixture and reliability jointly shape the rollout-level signal term in Eq. (6). We average Eq. (6) over $n$ steps in rollout $x$, let $q_{x,j} := q^*(\phi_{x,j})$ and the rollout-level $q$-dependent signal term takes the form

$$S(x) := \frac{1}{n}\sum_{j=1}^{n}(1 - 4\eta_{x,j})\,q_{x,j}(1 - q_{x,j})\,\|\phi_{x,j}\|^2.$$

where $\eta_{x,j}$ is the step-dependent noise rate, and can be

approximated by $\eta_{\text{eff}}(K_{x,j})$. The signal term $S(x)$ is determined by the multiplication of two factors: the term $q_{x,j}(1 - q_{x,j})$ favors rollouts with uncertain steps; while $(1 - 4\eta_{x,j})$ favors smaller label noise $\eta_{x,j}$ (larger $K_{x,j}$ or $s_{x,j}$ based on the monotonicity). Therefore, $S(x)$ of a rollout is large only when the multiplication of uncertainty and reliability is large. In practice, we approximate the teacher uncertainty $\theta_x := \bar{q}_x(1 - \bar{q}_x)$ using the observable label mixture $\hat{p}_x(1 - \hat{p}_x)$. Section 4 distills this into the Balanced-Information Score used in our data selection method.

# 4. Balanced-Information Score

Motivated by the empirical results above and the theoretical analysis, we introduce the Balanced-Information Score (BIS) as a rollout-level scoring mechanism that prioritizes informative rollouts for MPRM training.

**Setting.** Consider an MPRM training set where each rollout $x$ contains $n$ annotated steps with MC scores $\{s_j\}_{j=1}^{n} \subset [0,1]$. Following the standard binarization rule, each step is assigned a hard label $y_j = \mathbb{I}[s_j > 0]$.

**Rollout-level Quantities.** For a rollout $x$, define the positive-step ratio $p_{\text{pos}}(x) = \frac{1}{n}\sum_{j=1}^{n} y_j$, used to quantify label mixture, and a rollout-level reliability measure $R(x)$:

$$R(x) = \begin{cases} \dfrac{1}{n_{\text{pos}}}\sum_{j:y_j=1} s_j, & n_{\text{pos}} > 0, \\ 1, & n_{\text{pos}} = 0, \end{cases}$$

where $n_{\text{pos}} = \sum_{j=1}^{n} y_j$. Rollouts with $n_{\text{pos}} = 0$ contain no positive labeled steps to average over, so we fix $R(x) = 1$.

**Balanced-Information Score.** We define the Balanced-Information Score (BIS) of rollout $x$ as:

$$\text{BIS}(x) = \big(p_{\text{pos}}(x)\,(1 - p_{\text{pos}}(x)) + \alpha\big)\,R(x),$$

where the hyperparameter $\alpha > 0$ is a small smoothing constant that assigns a non-zero weight to low-mixture rollouts. The term $p_{\text{pos}}(1 - p_{\text{pos}})$ favors mixed rollouts that contain both correct and incorrect steps, while $R(x)$ favors rollouts whose positive steps are reliably correct under MC estimation. Therefore, $\text{BIS}(x)$ is highest for rollouts that provide both clear negative signals and trustworthy positive anchors.

**Subset Construction with Keep Ratio $\rho$.** Given a global keep-ratio $\rho \in (0,1)$ applied uniformly across all sources, we assign BIS to every MC-annotated rollout to build a data-efficient training set. Within each source dataset, we rank rollouts by $\text{BIS}(x)$ in descending order and keep the top $\rho$ fraction. We then concatenate the selected rollouts over all sources to form the BIS-selected subset. This procedure only relies on existing step-level MC scores and does not require additional supervision or extra model calls. The downstream MPRM can be trained from this $\rho$-subset with the same training setup of Section 2.1.

# 5. Experiments

## 5.1. Training Setup

We conduct experiments with two backbones: InternVL2.5-8B (Chen et al., 2024c) and Qwen2.5-VL-7B (Bai et al., 2025), and evaluate different methods on VisualProcessBench (Wang et al., 2025b). To study data efficiency, we sub-sample rollouts with keep ratios $\rho \in \{5, 10, 15, 25, 35, 50\}\%$ and compare BIS-$\rho$ against Random-$\rho$ under matched budgets. We additionally include heuristic subset baselines as ablations at selected budgets. All models are trained for a single pass over their retained rollouts, with training steps and learning-rate scheduling scaled proportionally to $\rho$. We also report Best-of-$N$ evaluation with MPRM reranking. Full training details are provided in Appendix C.

## 5.2. Main Results

**BIS recovers full-data performance at small ratios and consistently outperforms random sub-sampling.** Table 2 compares BIS-$\rho$ to Random-$\rho$ under identical rollout budgets. Across both backbones, BIS reaches full-data performance at small $\rho$ and remains consistently stronger than random sub-sampling, with the largest gains at small rollout budgets. For InternVL2.5-8B, BIS already matches the full-data performance at $\rho=10\%$, reaching an overall micro-F1 of 65.46%, a +2.6 points gain over the random baseline Random-10%, while using only one tenth of the rollouts and updates. For Qwen2.5-VL-7B, BIS shows even larger advantages in the extremely low-budget regime: it improves over random sub-sampling by +10.9 points at $\rho=5\%$ and +5.5 points at $\rho=15\%$, and already reaches the full-data reference at $\rho=25\%$. We report the complete training dynamics for all keep ratios in Appendix G, including both overall micro-F1 and per-source macro-F1, and show that BIS maintains clear advantages over random sub-sampling throughout training. BIS also shows a clear scaling trend with $\rho$: performance improves rapidly at small budgets, peaks at a moderate keep ratio, and can slightly drop afterwards, as increasing $\rho$ mainly adds lower-BIS rollouts that are less informative under our "mixture $\times$ reliability" criterion. Since $\rho=25\%$ performs strongly for both backbones, we use it for analysis in the following experiments.

**Effect of labeling scheme.** We further study the impact of the threshold for binarizing MC scores into hard labels and show the results in Table 2. First, training with soft labels is clearly inferior to default hard-label scheme. This is consistent with MC scores being noisy and coarsely discretized estimates, thus encouraging the MPRM to fit sampling noises. Second, increasing the binarization threshold to above 0 consistently degrades performance, suggesting that low-MC labels conflate hard cases with noisy pseudo-positives and that stricter thresholding can mislabel hard cases as nega-

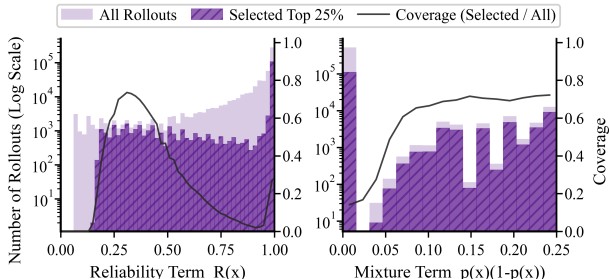

*Figure 3.* Distributions of the reliability term $R(x)$ (left) and mixture term $p(x)(1 - p(x))$ (right) on VisualPRM400K-v1.1, comparing all rollouts and the BIS-25% subset. The black curve shows the coverage (Selected / All).

tives. BIS therefore avoids this ambiguous low-MC regime by prioritizing mixed yet reliable rollouts.

**BIS improves best-of-$N$ reranking.** We further evaluate MPRM in a practical best-of-$N$ reranking setting with $N = 16$ candidates per problem on four different benchmarks (MM-K12 (Du et al., 2025), OlympiadBench (He et al., 2024), MathVerse (Zhang et al., 2024), and MathVista (Lu et al., 2024)) in Table 3. The full evaluation protocol is described in Appendix C. Consistent with the main results, MPRMs trained on BIS-selected subsets achieve the strongest best-of-$N$ performance across all benchmarks, outperforming both Random-25% and the full-data MPRM. This suggests that BIS yields robust improvements in MPRM effectiveness beyond benchmark-specific behavior.

**BIS favors moderate-$R(x)$ rollouts.** Figure 3 shows the distribution of the reliability term $R(x)$ on VisualPRM400K-v1.1 for all rollouts and the BIS-25% subset. Additional per-source statistics are provided in Appendix H. The black curve demonstrates the selected data coverage over the full data. BIS strongly suppresses the low-reliability rollouts where $R(x)$ are small which tend to include noisy step labels. Meanwhile, the coverage peaks at moderate $R(x)$ (around 0.2–0.6) and decreases as $R(x)$ becomes very large, showing that BIS does not simply maximize $R(x)$. This is because BIS jointly considers reliability and mixture: rollouts with large $R(x)$ may still have low mixture, resulting in a low overall BIS score. Consistently, the right panel shows that BIS favors rollouts with higher mixture, explaining why high-$R(x)$ rollouts are not always preferred.

**Ablation study: both mixture and reliability matter.** Table 4 ablates the two components of BIS under the same 25% rollout budget. Mixed-25% and Low-MC-25% are heuristic subsets of Section 2.3, while Reliable-25% retains the top 25% of rollouts ranked by $R(x)$. We observe that only using the mixture score (Mixed-25%) is competitive but still consistently weaker than the BIS both on average and on

*Table 2.* Overall micro-F1 and per-source macro-F1 on VisualProcessBench for full-data training and sub-sampled rollouts under different keep ratios $\rho$. "Soft" uses the raw MC scores as continuous soft targets; "Hard ($\tau$)" uses binary labels with threshold $\tau$. Bold numbers denote the column-wise maximum within each subset group. "Base" denotes the original backbone model without any additional training.

| Dataset | Overall | MathVision | MathVerse | MMMU | DynaMath | WeMath |
|---|---|---|---|---|---|---|
| *InternVL2.5-8B* | | | | | | |
| Base | 52.28 | 52.40 | 52.04 | 50.21 | 54.85 | 49.95 |
| *Full-Data (565k – 1,100 steps)* | | | | | | |
| Hard ($\tau$=0) | 65.12 | 65.77 | 65.43 | 61.84 | 66.17 | 63.56 |
| Hard ($\tau$=1/N) | 64.26 | 64.91 | 63.91 | 61.07 | 65.44 | 64.80 |
| Hard ($\tau$=2/N) | 63.02 | 61.61 | 62.92 | 59.70 | 65.83 | 65.09 |
| Soft | 61.54 | 60.78 | 60.47 | 62.05 | 62.91 | 64.38 |
| *5% Subsets (28k – 55 steps)* | | | | | | |
| Random-5% | 63.34 | 64.95 | 62.42 | 58.94 | **65.91** | 62.57 |
| BIS-5% | **64.51** | **66.66** | **64.53** | **60.30** | 63.80 | **64.40** |
| *10% Subsets (56k – 110 steps)* | | | | | | |
| Random-10% | 62.86 | 64.65 | 62.14 | 60.25 | 62.98 | 63.25 |
| BIS-10% | **65.46** | **66.90** | **65.07** | **63.35** | **65.56** | **65.10** |
| *15% Subsets (84k – 165 steps)* | | | | | | |
| Random-15% | 63.27 | 65.06 | 63.00 | 57.23 | 64.17 | 63.96 |
| BIS-15% | **64.98** | **67.09** | **64.44** | **61.80** | **64.58** | **65.40** |
| *25% Subsets (141k – 275 steps)* | | | | | | |
| Random-25% | 63.37 | 64.49 | 62.60 | 58.32 | **65.83** | 63.67 |
| BIS-25% | **65.46** | **67.98** | **64.86** | **60.49** | 65.72 | **65.59** |
| *35% Subsets (198k – 385 steps)* | | | | | | |
| Random-35% | 63.52 | 65.50 | 63.49 | 58.01 | 64.00 | 63.08 |
| BIS-35% | **64.98** | **67.25** | **64.47** | **59.61** | **65.79** | **64.82** |
| *50% Subsets (283k – 550 steps)* | | | | | | |
| Random-50% | 64.02 | 64.55 | **63.94** | 60.25 | 65.38 | 64.14 |
| BIS-50% | **65.00** | **65.84** | 63.79 | **63.03** | **66.66** | **66.03** |

| Dataset | Overall | MathVision | MathVerse | MMMU | DynaMath | WeMath |
|---|---|---|---|---|---|---|
| *Qwen2.5-VL-7B* | | | | | | |
| Base | 49.68 | 50.22 | 49.58 | 49.85 | 49.62 | 48.51 |
| *Full-Data (565k – 1,100 steps)* | | | | | | |
| Hard ($\tau$=0) | 65.57 | 66.05 | 65.29 | 63.23 | 66.24 | 66.40 |
| Hard ($\tau$=1/N) | 65.27 | 66.17 | 64.53 | 63.83 | 66.60 | 64.55 |
| Hard ($\tau$=2/N) | 62.72 | 62.33 | 62.18 | 62.56 | 63.37 | 64.65 |
| Soft | 62.23 | 62.17 | 61.25 | 61.44 | 62.88 | 65.55 |
| *5% Subsets (28k – 55 steps)* | | | | | | |
| Random-5% | 53.54 | 54.38 | 52.82 | 55.85 | 52.80 | 53.04 |
| BIS-5% | **64.42** | **66.69** | **64.20** | **62.72** | **63.34** | **63.12** |
| *10% Subsets (56k – 110 steps)* | | | | | | |
| Random-10% | 61.99 | 63.42 | 61.85 | 58.37 | 62.24 | 61.97 |
| BIS-10% | **64.63** | **66.05** | **64.14** | **61.33** | **64.64** | **66.05** |
| *15% Subsets (84k – 165 steps)* | | | | | | |
| Random-15% | 59.82 | 59.32 | 60.58 | 54.97 | 61.32 | 60.36 |
| BIS-15% | **65.29** | **66.50** | **64.03** | **66.40** | **64.80** | **66.58** |
| *25% Subsets (141k – 275 steps)* | | | | | | |
| Random-25% | 64.44 | 65.87 | 63.38 | 61.99 | **66.32** | 63.44 |
| BIS-25% | **65.53** | **66.84** | **64.48** | **63.60** | 66.19 | **66.66** |
| *35% Subsets (198k – 385 steps)* | | | | | | |
| Random-35% | 64.77 | 66.45 | 64.43 | 60.47 | 65.34 | 64.84 |
| BIS-35% | **65.69** | **66.77** | **65.50** | **63.37** | **65.65** | **65.99** |
| *50% Subsets (283k – 550 steps)* | | | | | | |
| Random-50% | 64.54 | 66.52 | 64.27 | 59.74 | **65.73** | 62.89 |
| BIS-50% | **65.02** | **67.22** | **64.30** | **60.80** | 65.62 | **64.99** |

*Table 3.* Best-of-$N$ evaluation on four benchmarks with MPRM reranking trained on different training sets.

| Model | MM-K12 | OlympiadBench | MathVerse | MathVista |
|---|---|---|---|---|
| InternVL2.5-8B | 33.13 | 8.65 | 35.31 | 52.77 |
| +MPRM$_{\text{Full-Data}}$ | 39.00 ↑ 5.87 | 12.00 ↑ 3.35 | 39.41 ↑ 4.10 | 57.50 ↑ 4.73 |
| +MPRM$_{\text{Random-25%}}$ | 39.40 ↑ 6.27 | 11.33 ↑ 2.68 | 39.41 ↑ 4.10 | 58.20 ↑ 5.43 |
| +MPRM$_{\text{BIS-25%}}$ | **41.00** ↑ 7.87 | **12.67** ↑ 4.02 | **40.89** ↑ 5.58 | **59.00** ↑ 6.23 |

*Table 4.* Ablations of BIS under a 25% rollout budget.

| Subset | Overall | MathVision | MathVerse | MMMU | DynaMath | WeMath |
|---|---|---|---|---|---|---|
| BIS-25% | **65.46** | **67.98** | **64.86** | 60.49 | **65.72** | **65.59** |
| Mixed-25% | 64.70 | 66.32 | 64.78 | 58.65 | 65.66 | 64.51 |
| △ | ↓0.76 | ↓1.66 | ↓0.08 | ↓1.84 | ↓0.06 | ↓1.08 |
| Reliable-25% | 62.75 | 62.12 | 63.14 | **60.52** | 63.85 | 63.14 |
| △ | ↓2.71 | ↓5.86 | ↓1.72 | ↑0.03 | ↓1.87 | ↓2.45 |
| Low-MC-25% | 64.18 | 66.31 | 64.31 | 59.40 | 64.16 | 62.97 |
| △ | ↓1.28 | ↓1.67 | ↓0.55 | ↓1.09 | ↓1.56 | ↓2.62 |

*Table 5.* Sensitivity of BIS to the smoothing constant $\alpha$ under a fixed 25% rollout budget.

| Subset | Overall | MathVision | MathVerse | MMMU | DynaMath | WeMath |
|---|---|---|---|---|---|---|
| $\alpha = 0.02$ | 64.84 | 66.63 | 64.69 | **60.49** | 64.78 | 65.27 |
| $\alpha = 0.05$ | **65.46** | **67.98** | **64.86** | **60.49** | **65.72** | **65.59** |
| $\alpha = 0.08$ | 64.86 | 67.10 | 64.47 | 59.96 | 64.88 | 65.35 |

pendix E, and further demonstrate that BIS-25% yields the highest or near-highest performance at almost all steps.

**BIS is robust to $\alpha$.** We ablate the smoothing constant $\alpha$ for the mixture term, which controls the lower-bound when $p_{\text{pos}}(1 - p_{\text{pos}})$ is small. Table 5 indicates that performance is broadly stable across different $\alpha$ values, with a consistent best choice at $\alpha = 0.05$. Overly small $\alpha$ can underweight low-mixture yet reliable trajectories and reduce the diversity of retained supervision, whereas overly large $\alpha$ weakens the mixture term and shifts selection toward reliability-only ranking. The intermediate value best balances these effects.

## 6. Conclusion

We study how to select Monte Carlo–annotated multimodal reasoning rollouts for training MPRMs. We found that randomly discarding most rollouts only mildly degrades performance, indicating that current training sets contain substantial redundancy. Our theoretical analysis explains that informative gradient updates concentrate on uncertain yet reliably labeled steps, while low-MC pseudo-positives mainly add variance. We propose the Balanced-Information

separate sources. Low-MC-25% shows a similar trend and remains weaker than BIS, indicating that heuristic filtering alone is insufficient to match BIS selection. In contrast, using only the reliability score (Reliable-25%) is clearly insufficient: it lags behind BIS on nearly all benchmarks and shows an advantage only on MMMU where performances are very close. This aligns with the analysis in Section 3.3 that the best performance is obtained only when both mixture and reliability are considered, where the mixture term provides contrast between positive and negative labels, and the reliability term steers away from noisy low-MC positive labels. We also plot the complete learning curves in Ap-

Score (BIS), which ranks rollouts by label mixture and reliability using only the MC signals already stored in the dataset. Empirical results demonstrate that BIS-selected subsets match or surpass full-data MPRM performance with as little as 10% of rollouts. Overall, our study provides a data-centric principle for curating future MPRM corpora.

## Impact Statement

This work aims to improve the data and compute efficiency of training multimodal process reward models. Successful adaptation of our method to practical model training will benefit the society by reducing energy cost.

## Acknowledgments

We would like to thank Han Li and Zhangchen Xu for their valuable insights. This material is based upon work supported in part by the National Science Foundation under Award No. 2541822, the NVIDIA Academic Grant Program, the WashU Ignite Interdisciplinary Grants, and the Ministry of Education, Singapore, under its Academic Research Fund (AcRF) Tier 1 grant (funded through the SUTD Assistant Professorship Scheme SAP 2025_001). Any opinions, findings and conclusions or recommendations expressed in this material are those of the author(s) and do not necessarily reflect the views of the funding agencies.

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

# Appendix

## Table of Contents

# A. Related Work

## A.1. Multimodal Process Reward Models under Automated Monte Carlo Process Supervision

Recent work (Zheng et al., 2025) shows that MPRMs improve multimodal reasoning both as dense rewards for reinforcement learning fine-tuning (Luo et al., 2025; Wang et al., 2025c; Liu et al., 2024; Fan et al., 2025) and as stepwise verifiers for inference-time trajectory ranking (Zhang et al., 2025a; Cao et al., 2025; Cao & Xie, 2025; Wang et al., 2025a; Tu et al., 2025; Hu et al., 2025). Unlike outcome rewards that score only the final answer (Lightman et al., 2024; Zhang et al., 2025b;f;e; Wang et al., 2024a), an MPRM (Chen et al., 2025; Ong et al., 2025; Kuang et al., 2025) provides dense, step-level supervision by mapping each intermediate multimodal reasoning state to a real-valued "on-track" score conditioned on the input images and text. Most standard MPRM corpora are built from Monte Carlo (MC) estimates computed on reasoning prefixes, with VisualPRM400K (Wang et al., 2025b) as a representative example. One common approach samples multiple continuations from each prefix and uses the empirical success rate to score step correctness (Wang et al., 2024d). A complementary line replaces plain sampling with structured search via Monte Carlo Tree Search, improving the stability of error localization and supervision signals (Luo et al., 2024; Wang et al., 2024b; Chen et al., 2024a). Despite their differences, MC-based annotators are inherently noisy under finite sampling and long-horizon multimodal reasoning, yielding unstable labels and low-success "pseudo-positives" that tightening the binarization threshold cannot simply fix and can even hurt MPRM performance (Wang et al., 2025b). Taken together, MC-annotated supervision is plentiful but highly uneven in information content, making rollout-level prioritization crucial for efficient and stable MPRM training.

## A.2. Data-efficient Process Supervision

Recent work on data-efficient process supervision for PRMs can be broadly grouped into three complementary categories. The first line of work optimizes the annotation pipeline itself (Han et al., 2025; Sun et al., 2025; Zhang et al., 2025d; Wang et al., 2025a; Zhang et al., 2025g). These methods improve Monte Carlo or search-based annotators with techniques such as Monte Carlo tree search, tool grounding, and consensus-based filtering, enabling each trajectory to yield higher-quality step labels from fewer or cheaper model calls. A second line of work focuses on learning robust process supervision from weak, noisy, or indirect feedback (Ding et al., 2025; Xiong et al., 2025; Khalifa et al., 2025; Cui et al., 2025; Chen et al., 2024a). These approaches design objectives and model forms that allow PRMs to learn effectively from imperfect MC labels and, in some cases, even from outcome-only signals, thereby reducing reliance on expensive, high-quality process annotations. Our work belongs to a third line that focuses on data selection and supervision allocation. DreamPRM (Cao et al., 2025) and DreamPRM-1.5 (Cao & Xie, 2025) adjust dataset and example weights via bi-level optimization, while ACTPRM (Duan et al., 2025) and SCAN (Ding et al., 2025) select which samples to query or refine with expensive Monte Carlo estimation under a limited annotation budget. In contrast, our rollout-level BIS is a scalar score computed post hoc from existing MC statistics, enabling data selection with no extra model calls, relabeling, or changes to the underlying PRM architecture or training objective.

# B. VisualProcessBench Statistics

Table 6 summarizes the composition of VisualProcessBench (Wang et al., 2025b), including the number of problems per source dataset, the distribution of source solutions across base models, the breakdown of correct/incorrect/neutral steps, and basic length statistics. In total, the benchmark contains 2,866 solution trajectories with 26,950 annotated steps, providing a reasonably large and diverse testbed for step-level evaluation.

*Table 6.* Statistics of VisualProcessBench (Wang et al., 2025b).

| Item | Value |
|---|---|
| *Dataset Composition* | |
| Total Samples | 2866 |
| – MMMU (Yue et al., 2024) | 267 |
| – MathVision (Wang et al., 2024c) | 712 |
| – MathVerse (Zhang et al., 2024) | 1026 |
| – DynaMath (Zou et al., 2025) | 570 |
| – WeMath (Qiao et al., 2025) | 291 |
| *Source Solutions* | |
| Source Solutions | 2866 |
| – GPT-4o (OpenAI, 2024) | 870 |
| – Claude-3.5-Sonnet (Anthropic, 2024) | 865 |
| – QvQ-72B-Preview (Qwen Team, 2024) | 825 |
| – InternVL2.5-78B (Chen et al., 2024c) | 306 |
| *Steps* | |
| Total Steps | 26950 |
| – Correct Steps | 16585 |
| – Incorrect Steps | 7691 |
| – Neutral Steps | 2674 |
| *Length Statistics* | |
| Query Word Length Quartile | (22, 24, 50) |
| Response Word Length Quartile | (137, 193, 552) |
| Step Word Length Quartile | (13, 31, 67) |
| Number of Steps per Solution | 9.4 |

# C. Experimental Setup and Implementation Details

This appendix provides the full experimental setup omitted from the main text for brevity, including the model and training data, the MPRM training objective, optimization and hardware settings, and the evaluation protocol.

*Table 7.* Training, model, and hardware hyperparameters (shared across all data-selection conditions).

| Item | Value |
|------|-------|
| *Optimization* | |
| Optimizer | AdamW |
| Learning Rate | $1 \times 10^{-5}$ |
| Weight Decay | 0.05 |
| AdamW Betas | $(0.9, 0.999)$ |
| AdamW $\epsilon$ | $1 \times 10^{-8}$ |
| LR Schedule | Linear Warmup + Cosine Decay |
| Warmup Ratio | 0.05 |
| Gradient Clipping | Enabled via DeepSpeed (`gradient_clipping=auto`, using the Trainer default `max_grad_norm`) |
| Precision | `bf16` |
| *Batching and Training Budget* | |
| Per-device Batch Size | 2 |
| Gradient Accumulation Steps | 64 |
| Global Batch Size | $B = 512$ |
| Epochs | 1 (single pass; default for all experiments unless otherwise noted) |
| Max Sequence Length | 8192 (truncate from the end) |
| Optimization Steps | $T = \lceil N/B \rceil$ for a pool of $N$ rollouts |
| *Model, Input, and Hardware* | |
| Backbone | InternVL2.5-8B & Qwen2.5-VL-7B |
| Trainable Modules | LLM + multimodal fusion MLP (vision backbone frozen) |
| Image Size | InternVL2.5-8B: 448, dynamic resolution enabled; max 6 patches. Qwen2.5-VL-7B: dynamic resizing (min/max=784/200704). |
| GPUs | $4 \times$ NVIDIA H100 80GB |

**Model and Training Data**  We use InternVL2.5-8B (Chen et al., 2024c) as the default backbone, following prior MPRM work (Wang et al., 2025b; Du et al., 2025). In the main experiments, we additionally report results with a second backbone, Qwen2.5-VL-7B (Bai et al., 2025). For both models, we freeze the vision encoder and fine-tune the language model together with the multimodal projector modules. We use the default vision setup and input preprocessing of each backbone; the corresponding model and input hyperparameters are summarized in Table 7. We train on VisualPRM400K-v1.1[1] (Wang et al., 2025b), choosing v1.1 because it exposes per-step MC scores, whereas the v1 release only provides binarized labels. This dataset was sampled from 38 different data sources. Detailed training-data statistics are reported in Table 8.

**MPRM Training Objective**  Each training example consists of the question, the associated images, and a step-by-step solution, where every step is followed by a special token <prm>. The tokenizer inserts this special token into the text stream and the data loader attaches the corresponding binary label to its position. During training, the model is supervised only on these <prm> positions: the logits at each placeholder are restricted to the two reward tokens (``Yes'', ``No'') and optimized with a two-way cross-entropy loss, so that the probability of ``Yes'' serves as the score for that step.

**Optimization and Implementation**  Table 7 summarizes the hyperparameters and implementation details shared by all experiments. The learning rate is linearly warmed up over the first $5\%$ of optimization steps and then cosine-decayed to zero over the remaining steps. Under our default single-pass protocol (i.e., one epoch over the selected training pool), the total number of optimization steps is recomputed for each data regime as $T = \lceil N/B \rceil$, where $N$ is the number of rollouts in the

---

[1] https://huggingface.co/datasets/OpenGVLab/VisualPRM400K-v1.1-Raw

pool and $B$ is the global batch size. Training is implemented in PyTorch (Paszke et al., 2019) with HuggingFace `Trainer` from Transformers (Wolf et al., 2020) and DeepSpeed (Rasley et al., 2020) ZeRO-3 for memory efficiency.

**Evaluation Protocol**   For each VisualProcessBench (Wang et al., 2025b) instance, we concatenate the question with the provided step-by-step rationale and insert `<prm>` after every step, mirroring training. The model produces a scalar score per step (the ``Yes'' probability at the corresponding placeholder). Given a threshold $\tau$, we classify steps with scores $\geq \tau$ as positive and those $< \tau$ as negative, ignoring neutral labels. Following the benchmark protocol, we select a single global threshold per model on a held-out split by sweeping $\tau$ and maximizing the micro-averaged F1 across all sources; we then report the overall micro-F1 in the main text and provide per-source macro-F1 breakdowns in the appendix.

**Best-of-$N$ Reranking Protocol**   We report best-of-$N$ reranking results with $N = 16$ for all benchmarks in Table 3. For each problem, we first sample 16 candidate step-by-step rollouts using InternVL2.5-8B with standard stochastic decoding (temperature $= 0.7$, top-$p = 0.9$, top-$k = 30$, and max_new_tokens $= 2048$). Each candidate is formatted as a sequence of reasoning steps followed by a final answer. To rerank candidates, we apply the MPRM to obtain a scalar score at the step level. Given a candidate rollout $\tau$ with $T$ reasoning steps and step scores $\{s_t\}_{t=1}^{T}$, we compute a trajectory-level score by averaging over steps:

$$S(\tau) = \frac{1}{T} \sum_{t=1}^{T} s_t.$$

We then select the candidate with the highest $S(\tau)$ as the final prediction for evaluation.

## D. Extended Results for Random Sub-sampling (Section 2.2)

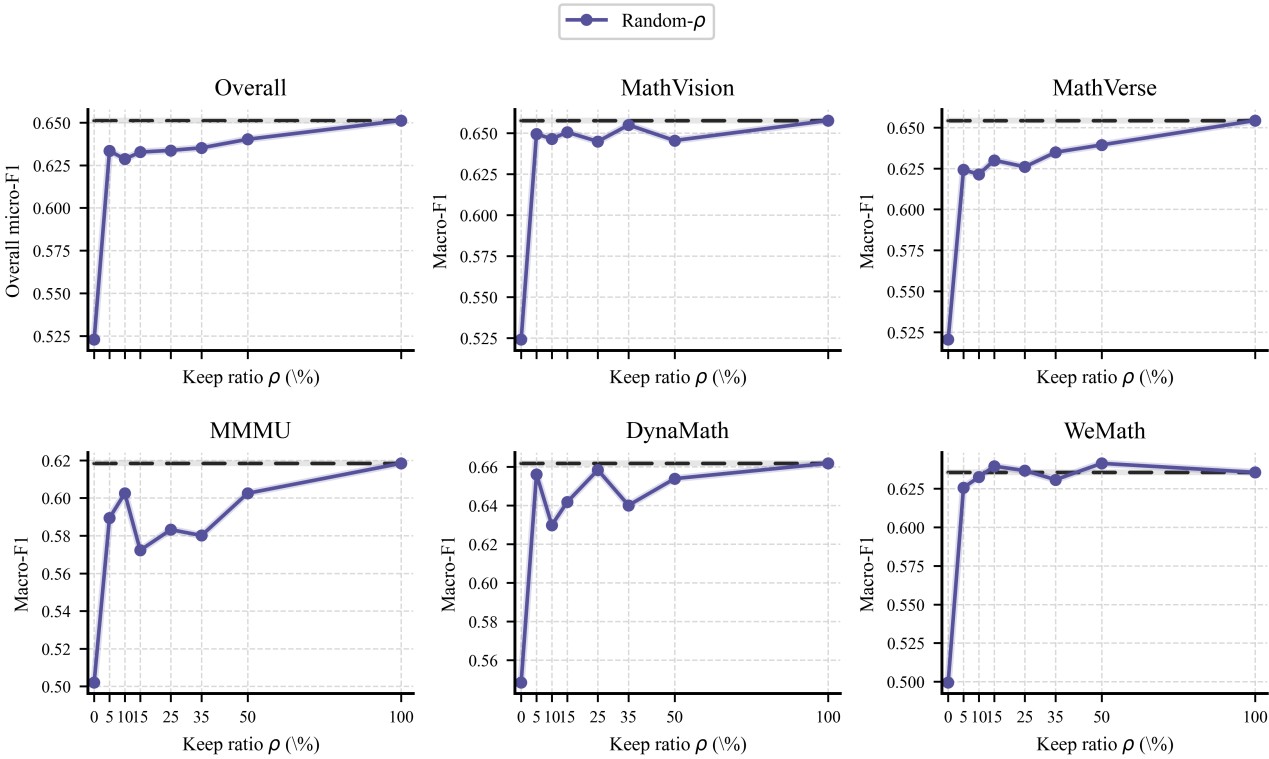

*Figure 4.* Single-pass scaling with random sub-sampling (Random-$\rho$) on VisualProcessBench. The top-left panel reports Overall micro-F1; the remaining panels show macro-F1 on each source dataset. The dashed horizontal line marks the Full-Data ($\rho = 100\%$) model.

Figure 4 extends the single-pass scaling plot in Figure 2a by breaking down the Random-$\rho$ behavior on VisualProcessBench by source. The top-left panel reproduces the Overall micro-F1 curve, while the remaining panels show macro-F1 on each benchmark. Across sources, performance rises sharply when $\rho$ increases from 0 to a small fraction (e.g., 5%), and then quickly saturates: further enlarging the random pool beyond the low two-digit range yields only mild additional gains. This per-source view mirrors the redundancy-dominated scaling discussed in Section 2.2.

Figure 5 focuses on the $\rho = 25\%$ working point and compares Full-Data and Random-25% under a matched update budget. The top-left panel shows Overall micro-F1, and the remaining panels report macro-F1 on each VisualProcessBench source. Across sources, the full-data run has a systematic but moderate edge over the Random-25% subset, and the gap is smaller than one might expect after discarding 75% of the rollouts. This pattern is consistent with a regime in which additional rollouts yield diminishing returns.

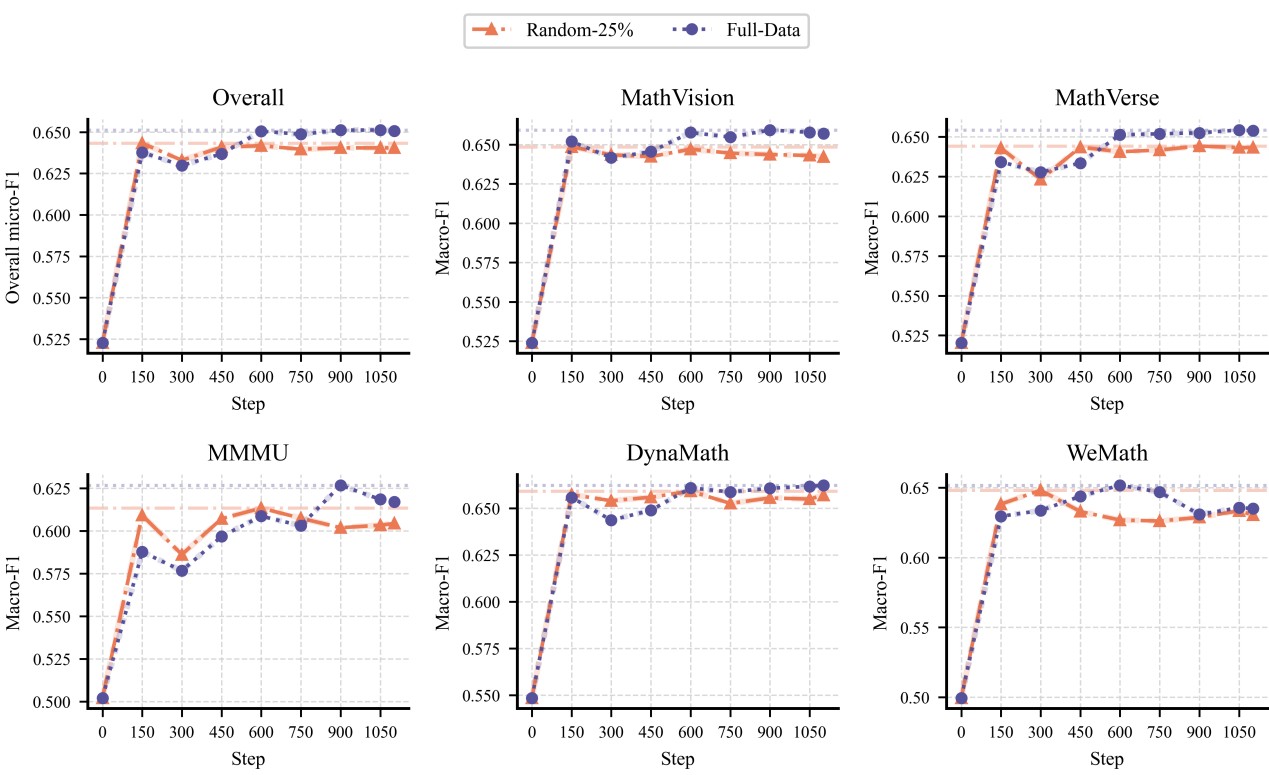

*Figure 5.* VisualProcessBench performance vs. training step when training on the Full-Data and Random-25% settings of VisualPRM400K-v1.1. The top-left panel shows the Overall micro-F1 aggregated over all sources, while the remaining panels show macro-F1 on each individual VisualProcessBench source.

# E. Extended Results for 25% Subsets (Sections 2.3 and 5.2)

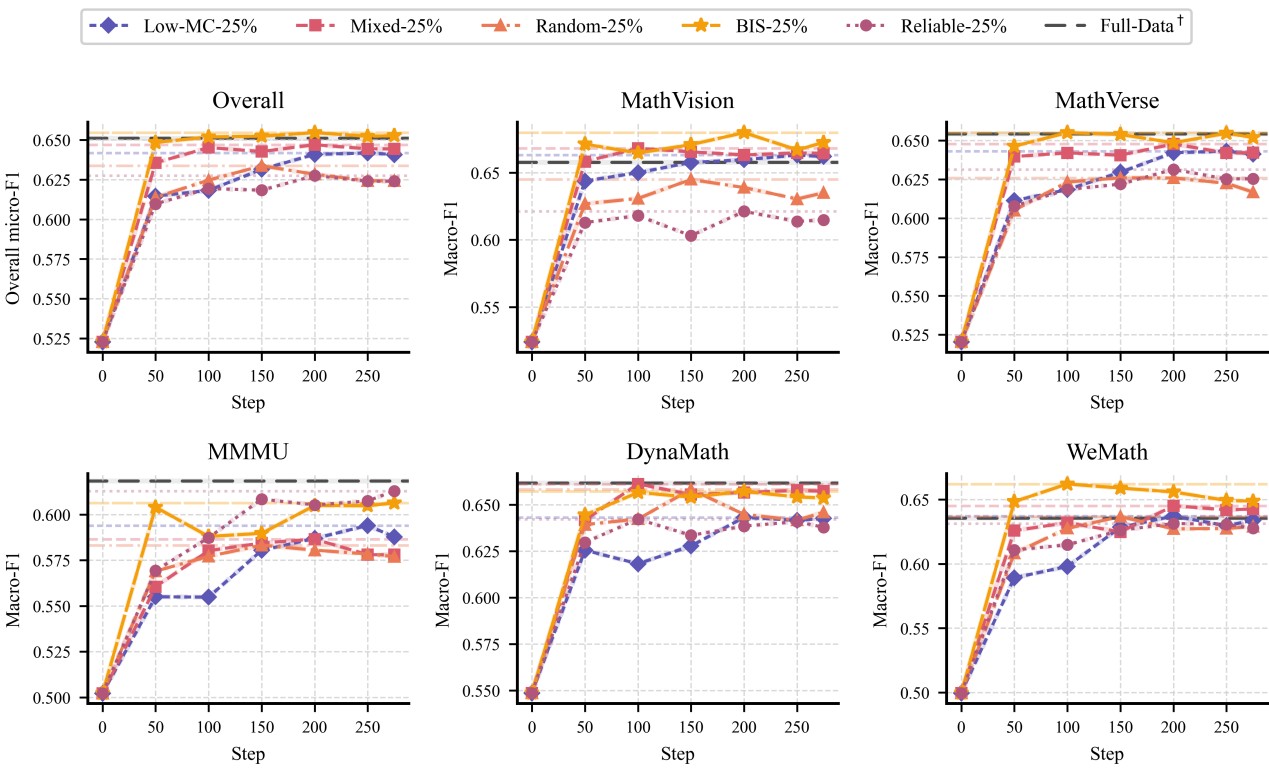

*Figure 6.* Evaluation performance vs. training step on VisualProcessBench for four 25% subsets of VisualPRM400K-v1.1. The top-left panel shows overall micro-F1; the remaining panels show macro-F1 on each source dataset. All 25% subset models are trained for a single pass over their respective training pools. Full-Data[†] shows the best checkpoint from a one-epoch Full-Data run (4× more optimization steps), shown only as a reference. BIS-25% consistently outperforms other subsets on overall and on most individual sources across the training trajectory, not only at a single checkpoint.

Table 4 in the main text reports overall micro-F1 and per-source macro-F1 on VisualProcessBench under the same 25% rollout budget, comparing BIS-25% against three baselines: Mixed-25%, Reliable-25%, and Low-MC-25%. Here we complement Table 4 with the full training curves of these 25% subsets on VisualProcessBench, and additionally include Random-25% as a standard sub-sampling baseline. Figure 6 plots overall micro-F1 and per-source macro-F1 as a function of training step. These curves provide a dynamic view of how BIS re-allocates the fixed update budget compared with Random-25%, Low-MC-25%, Mixed-25%, and Reliable-25%. Across sources, BIS-25% yields the highest or near-highest curve at almost all steps. Combined with the aggregate scores in Table 4, these extended results corroborate that BIS is strictly more effective than these 25% baselines under the same rollout and update budget.

# F. Theoretical Details

## F.1. A Canonical Logistic Case for the Scaling Decomposition

This section makes the decomposition in Eq. (4) precise in the logistic teacher–student setting of Section 3.1, and derives the $\mathcal{O}(N_{\text{eff}}^{-1/2})$ data term and $\mathcal{O}(T^{-1/2})$ optimization term step by step.

**Setup.** Each training step is a pair $(\phi, Y)$ with $\phi \in \mathbb{R}^d$ and $Y \in \{0, 1\}$. The population logistic loss is

$$\mathcal{L}(w) = \mathbb{E}_{(\phi, Y)}\big[-Y \log q_w(\phi) - (1 - Y) \log\big(1 - q_w(\phi)\big)\big], \quad q_w(\phi) = \sigma(\langle w, \phi \rangle),$$

and $w^*$ denotes its minimizer. Let $N_{\text{eff}}$ denote the number of i.i.d. training steps after thinning the pool, and

$$\mathcal{L}_{N_{\text{eff}}}(w) = \frac{1}{N_{\text{eff}}} \sum_{i=1}^{N_{\text{eff}}} \ell(w; \phi_i, Y_i), \quad \ell(w; \phi, Y) = -Y \log q_w(\phi) - (1 - Y) \log\big(1 - q_w(\phi)\big)$$

be the empirical logistic loss. We write

$$\widehat{w}_{N_{\text{eff}}} \in \arg\min_{w \in \mathcal{W}} \mathcal{L}_{N_{\text{eff}}}(w)$$

for an empirical minimizer, and $w_T$ for the SGD iterate after $T$ updates on this finite sample.

**Assumptions.** We assume:

(A1) (*Bounded features*) There exists $B > 0$ such that $\|\phi\|_2 \leq B$ almost surely.

(A2) (*Well-specified logistic teacher*) There exists $w^* \in \mathbb{R}^d$ such that $\Pr(Y = 1 \mid \phi) = \sigma(\langle w^*, \phi \rangle)$ almost surely.

(A3) (*Strong convexity and smoothness on a bounded domain*) Assume $\mathcal{L}$ is $\mu$-strongly convex and $L$-smooth on a closed, convex, bounded set $\mathcal{W} \subset \mathbb{R}^d$ containing $w^*$, with $\sup_{w \in \mathcal{W}} \|w\|_2 \leq R$. Such a condition can be obtained, for example, by adding a small $\ell_2$ penalty on $w$.

(A4) (*SGD with decaying steps*) We run projected SGD so that $w_t \in \mathcal{W}$ for all $t$. Stochastic gradients are computed on i.i.d. samples and have bounded second moment, and the step sizes satisfy $\eta_t = \eta_0/\sqrt{t}$ with $\eta_0$ small enough so that $\eta_t \leq 1/L$.

**Goal.** We bound the excess population loss

$$\mathbb{E}\big[\mathcal{L}(w_T)\big] - \mathcal{L}(w^*)$$

where the expectation is over both the draw of the training set and the randomness of SGD, and show that it decomposes into a $\mathcal{O}(N_{\text{eff}}^{-1/2})$ data term and a $\mathcal{O}(T^{-1/2})$ optimization term.

**Step 1: Decomposition into data and optimization terms.** Insert and subtract $\widehat{w}_{N_{\text{eff}}}$:

$$\mathbb{E}\big[\mathcal{L}(w_T)\big] - \mathcal{L}(w^*) = \mathbb{E}\big[\mathcal{L}(w_T) - \mathcal{L}(\widehat{w}_{N_{\text{eff}}})\big] + \mathbb{E}\big[\mathcal{L}(\widehat{w}_{N_{\text{eff}}}) - \mathcal{L}(w^*)\big]$$
$$=: \text{Opt}(T, N_{\text{eff}}) + \text{Data}(N_{\text{eff}}). \tag{11}$$

The first term measures optimization error after $T$ SGD updates on a fixed finite sample; the second term measures the gap between the empirical and population optima due to finite data.

**Step 2: Bounding the finite-data term.** By definition,

$$\text{Data}(N_{\text{eff}}) = \mathbb{E}\big[\mathcal{L}(\widehat{w}_{N_{\text{eff}}}) - \mathcal{L}(w^*)\big].$$

Using the standard optimism of empirical risk minimization (ERM) argument,

$$\mathcal{L}(\widehat{w}_{N_{\text{eff}}}) - \mathcal{L}(w^*) = \big(\mathcal{L}(\widehat{w}_{N_{\text{eff}}}) - \mathcal{L}_{N_{\text{eff}}}(\widehat{w}_{N_{\text{eff}}})\big) + \big(\mathcal{L}_{N_{\text{eff}}}(\widehat{w}_{N_{\text{eff}}}) - \mathcal{L}_{N_{\text{eff}}}(w^*)\big) + \big(\mathcal{L}_{N_{\text{eff}}}(w^*) - \mathcal{L}(w^*)\big)$$
$$\leq \big(\mathcal{L}(\widehat{w}_{N_{\text{eff}}}) - \mathcal{L}_{N_{\text{eff}}}(\widehat{w}_{N_{\text{eff}}})\big) + \big(\mathcal{L}_{N_{\text{eff}}}(w^*) - \mathcal{L}(w^*)\big)$$
$$\leq 2 \sup_{w \in \mathcal{W}} \big|\mathcal{L}(w) - \mathcal{L}_{N_{\text{eff}}}(w)\big|.$$

Taking expectations and applying uniform convergence then yields

$$\text{Data}(N_{\text{eff}}) \leq 2\mathbb{E}\left[\sup_{w \in \mathcal{W}} \left|\mathcal{L}(w) - \mathcal{L}_{N_{\text{eff}}}(w)\right|\right]. \tag{12}$$

Under (A1)–(A3), the logistic loss $\ell(w; \phi, Y)$ is Lipschitz in $w$ on $\mathcal{W}$ and the class $\{\ell(w; \cdot, \cdot) : w \in \mathcal{W}\}$ has bounded Rademacher complexity. Standard uniform convergence bounds for Lipschitz losses in generalized linear models (Shalev-Shwartz & Ben-David, 2014) then imply the existence of a constant $C_{\text{data}} > 0$ such that

$$\mathbb{E}\left[\sup_{w \in \mathcal{W}} \left|\mathcal{L}(w) - \mathcal{L}_{N_{\text{eff}}}(w)\right|\right] \leq \frac{C_{\text{data}}}{\sqrt{N_{\text{eff}}}}. \tag{13}$$

Combining (12) and (13) and absorbing the factor 2 into the constant gives

$$\text{Data}(N_{\text{eff}}) \leq \frac{C_{\text{data}}}{\sqrt{N_{\text{eff}}}}. \tag{14}$$

Here $C_{\text{data}}$ depends on the feature bound $B$ and the domain radius $R$ (and thus grows with $BR$).

**Step 3: Bounding the optimization term.** We now control the optimization error $\text{Opt}(T, N_{\text{eff}}) = \mathbb{E}[\mathcal{L}(w_T) - \mathcal{L}(\widehat{w}_{N_{\text{eff}}})]$. Conditioned on the fixed sample $\{(\phi_i, Y_i)\}_{i=1}^{N_{\text{eff}}}$, let

$$F(w) := \mathcal{L}_{N_{\text{eff}}}(w) = \frac{1}{N_{\text{eff}}} \sum_{i=1}^{N_{\text{eff}}} \ell(w; \phi_i, Y_i), \qquad \widehat{w}_{N_{\text{eff}}} \in \arg\min_{w \in \mathcal{W}} F(w).$$

We assume $F$ is $\mu$-strongly convex and $L$-smooth on $\mathcal{W}$. Moreover, we assume that the empirical minimizer $\widehat{w}_{N_{\text{eff}}}$ lies in the interior of $\mathcal{W}$, so that $\nabla F(\widehat{w}_{N_{\text{eff}}}) = 0$. The stochastic gradients $g_t$ used by SGD satisfy $\mathbb{E}[g_t \mid w_t] = \nabla F(w_t)$ and $\mathbb{E}[\|g_t\|^2 \mid w_t] \leq G^2$ for some $G > 0$. The SGD recursion on the empirical loss is

$$w_{t+1} = \Pi_{\mathcal{W}}(w_t - \eta_t g_t), \qquad \eta_t = \eta_0 t^{-1/2},$$

with $\eta_0$ small enough so that $\eta_t \leq 1/L$ for all $t$.

Define the mean squared distance to the empirical minimizer as

$$D_t := \mathbb{E}\left[\|w_t - \widehat{w}_{N_{\text{eff}}}\|^2\right].$$

A standard one-step expansion of $\|w_{t+1} - \widehat{w}_{N_{\text{eff}}}\|^2$, combined with the $\mu$-strong convexity and $L$-smoothness of $F$ and the bounded-variance assumption on $g_t$, implies that the sequence $(D_t)$ satisfies a recursion of the form

$$D_{t+1} \leq \left(1 - 2\mu\eta_t + 2L^2\eta_t^2\right) D_t + 2G^2\eta_t^2,$$

see Moulines & Bach (2011) for a detailed derivation. Since $\eta_t$ is non-increasing and $w_t \in \mathcal{W}$ for all $t$, and $\mathcal{W}$ is bounded so that $D_t \leq \text{diam}(\mathcal{W})^2$, we may absorb the $2L^2\eta_t^2 D_t$ term and $2\sigma^2$ into a single constant $G^2$, yielding the simplified recursion

$$D_{t+1} \leq (1 - 2\mu\eta_t) D_t + \eta_t^2 G^2. \tag{15}$$

Specializing to the step-size schedule $\eta_t = \eta_0 t^{-1/2}$, (15) can be rewritten as

$$D_{t+1} \leq \left(1 - \frac{c}{\sqrt{t}}\right) D_t + \frac{C}{t}, \tag{16}$$

for some constants $c, C > 0$ depending only on $\mu, \eta_0, G$.

**Lemma 1** (One-dimensional SGD recursion). *Let $(D_t)_{t \geq 1}$ be a nonnegative sequence satisfying (16) for all $t \geq 1$, with $c, C > 0$. Then there exists a constant $C' > 0$, depending only on $c, C$ and $D_1$, such that*

$$D_T \leq \frac{C'}{\sqrt{T}} \qquad \text{for all } T \geq 1.$$

This lemma is a direct corollary of standard results for stochastic approximation recursions; see, e.g., the standard mean-square error recursion and the corresponding non-asymptotic bound in Moulines & Bach (2011) (with $\alpha = 1/2$) for an explicit derivation of the $\mathcal{O}(T^{-1/2})$ rate. Applying Lemma 1 to (16) yields

$$D_T = \mathbb{E}\big[\|w_T - \widehat{w}_{N_{\text{eff}}}\|^2\big] \ \leq \ \frac{C'}{\sqrt{T}}. \tag{17}$$

By $L$-smoothness of $F$ (see Lemma 1.2.3 in Nesterov et al. (2018)), we have for any $w$

$$F(w) \ \leq \ F(\widehat{w}_{N_{\text{eff}}}) + \big\langle \nabla F(\widehat{w}_{N_{\text{eff}}}), \, w - \widehat{w}_{N_{\text{eff}}} \big\rangle + \frac{L}{2} \, \|w - \widehat{w}_{N_{\text{eff}}}\|^2.$$

Since $\widehat{w}_{N_{\text{eff}}}$ is a minimizer of $F$, $\nabla F(\widehat{w}_{N_{\text{eff}}}) = 0$, and thus

$$F(w_T) - F(\widehat{w}_{N_{\text{eff}}}) \ \leq \ \frac{L}{2} \, \|w_T - \widehat{w}_{N_{\text{eff}}}\|^2.$$

Taking expectations and combining with (17) gives

$$\mathbb{E}\big[\mathcal{L}_{N_{\text{eff}}}(w_T) - \mathcal{L}_{N_{\text{eff}}}(\widehat{w}_{N_{\text{eff}}})\big] \ \leq \ \frac{C_{\text{opt}}}{\sqrt{T}}, \tag{18}$$

for some constant $C_{\text{opt}} > 0$ depending only on $\mu, L, G$ and the initialization. In particular, higher Monte Carlo noise typically increases the second-moment bound $G^2$, which increases $C_{\text{opt}}$ and can make the optimization term dominant in a noise-limited regime.

To relate this bound on the empirical loss to the population loss, we insert and subtract $\mathcal{L}_{N_{\text{eff}}}$ and decompose

$$\mathbb{E}\big[\mathcal{L}(w_T) - \mathcal{L}(\widehat{w}_{N_{\text{eff}}})\big] = \mathbb{E}\big[\mathcal{L}_{N_{\text{eff}}}(w_T) - \mathcal{L}_{N_{\text{eff}}}(\widehat{w}_{N_{\text{eff}}})\big] + \mathbb{E}\big[(\mathcal{L}(w_T) - \mathcal{L}_{N_{\text{eff}}}(w_T)) - (\mathcal{L}(\widehat{w}_{N_{\text{eff}}}) - \mathcal{L}_{N_{\text{eff}}}(\widehat{w}_{N_{\text{eff}}}))\big]. \tag{19}$$

By the triangle inequality,

$$\big|\mathbb{E}\big[(\mathcal{L}(w_T) - \mathcal{L}_{N_{\text{eff}}}(w_T)) - (\mathcal{L}(\widehat{w}_{N_{\text{eff}}}) - \mathcal{L}_{N_{\text{eff}}}(\widehat{w}_{N_{\text{eff}}}))\big]\big| \ \leq \ 2\,\mathbb{E}\Big[\sup_{w \in \mathcal{W}} \big|\mathcal{L}(w) - \mathcal{L}_{N_{\text{eff}}}(w)\big|\Big]. \tag{20}$$

Applying the uniform-convergence bound (13) to (20) and combining with (18) and (19) yields

$$\text{Opt}(T, N_{\text{eff}}) = \mathbb{E}\big[\mathcal{L}(w_T) - \mathcal{L}(\widehat{w}_{N_{\text{eff}}})\big] \ \leq \ \frac{C_{\text{opt}}}{\sqrt{T}} + \frac{C'_{\text{data}}}{\sqrt{N_{\text{eff}}}}, \tag{21}$$

for some constant $C'_{\text{data}} > 0$. When we combine (21) with the finite-sample term in (14), the two $\mathcal{O}(N_{\text{eff}}^{-1/2})$ contributions can be aggregated into a single constant, leading to the overall decomposition in (4).

**Step 4: Putting the pieces together.** Substituting (14) and (21) into the decomposition (11) yields

$$\mathbb{E}\big[\mathcal{L}(w_T)\big] - \mathcal{L}(w^*) \ = \ \text{Opt}(T, N_{\text{eff}}) + \text{Data}(N_{\text{eff}}) \ \leq \ \frac{C_{\text{data}}}{\sqrt{N_{\text{eff}}}} + \frac{C_{\text{opt}}}{\sqrt{T}}.$$

These $\mathcal{O}(N_{\text{eff}}^{-1/2})$ and $\mathcal{O}(T^{-1/2})$ rates are conservative but sufficient for the scaling decomposition used in Section 3.2.

### F.2. Step-wise Gradient Variance

We derive the step-wise gradient variance expression used in Equation (5). Recall the teacher–student setup in Section 3.1. For a step with representation $\phi \in \mathbb{R}^d$ and clean label $Y \in \{0, 1\}$, the logistic-loss gradient at parameter $w$ is

$$g(\phi, Y; w) = \big(q_w(\phi) - Y\big)\,\phi, \qquad q_w(\phi) = \sigma(\langle w, \phi \rangle).$$

At the teacher parameter $w^*$, write

$$q^*(\phi) := q_{w^*}(\phi), \qquad Y \mid \phi \sim \text{Bernoulli}\big(q^*(\phi)\big).$$

Then

$$\mathbb{E}\big[Y \mid \phi\big] = q^*(\phi), \qquad \mathbb{E}\big[Y^2 \mid \phi\big] = q^*(\phi).$$

The conditional mean of the gradient at $w^*$ is

$$\mathbb{E}\big[g(\phi, Y; w^*) \mid \phi\big] = \big(q^*(\phi) - \mathbb{E}[Y \mid \phi]\big)\phi = 0,$$

so the conditional second moment equals the conditional variance:

$$
\begin{aligned}
\mathbb{E}\big[\|g(\phi, Y; w^*)\|^2 \mid \phi\big] &= \mathbb{E}\big[(q^*(\phi) - Y)^2 \mid \phi\big]\,\|\phi\|^2 \\
&= \Big(q^{*2}(\phi) - 2q^*(\phi)\,\mathbb{E}[Y \mid \phi] + \mathbb{E}[Y^2 \mid \phi]\Big)\,\|\phi\|^2 \\
&= \big(q^*(\phi) - q^{*2}(\phi)\big)\,\|\phi\|^2 \\
&= q^*(\phi)\big(1 - q^*(\phi)\big)\,\|\phi\|^2,
\end{aligned}
$$

which is exactly the expression in Equation (5).

## F.3. Symmetric Label Noise

We now derive the noisy-gradient expression used in Equation (6). Fix $\phi \in \mathbb{R}^d$ and write $q^*(\phi) = q_{w^*}(\phi)$. For brevity, set

$$q := q^*(\phi) \in (0, 1).$$

Let the clean label $Y \mid \phi \sim \mathrm{Bernoulli}(q)$ be flipped independently with probability $\eta \in [0, 1/2)$ to form a noisy label $\tilde{Y}$. The noisy gradient at $w^*$ is

$$\tilde{g}(\phi, \tilde{Y}; w^*) = (q - \tilde{Y})\,\phi.$$

Conditioned on $\phi$, we can express the distribution of the noisy label $\tilde{Y}$ by conditioning on the clean label $Y$ and applying the law of total probability:

$$
\begin{aligned}
\Pr(\tilde{Y} = 1 \mid \phi) &= \Pr(\tilde{Y} = 1, Y = 1 \mid \phi) + \Pr(\tilde{Y} = 1, Y = 0 \mid \phi) \\
&= \Pr(\tilde{Y} = 1 \mid Y = 1, \phi)\,\Pr(Y = 1 \mid \phi) + \Pr(\tilde{Y} = 1 \mid Y = 0, \phi)\,\Pr(Y = 0 \mid \phi) \\
&= (1 - \eta)\,q + \eta\,(1 - q) \\
&= q(1 - 2\eta) + \eta =: p_1,
\end{aligned}
$$

where $q = q^*(\phi)$ and $Y \mid \phi \sim \mathrm{Bernoulli}(q)$. Thus

$$\Pr(\tilde{Y} = 0 \mid \phi) = 1 - p_1 =: p_0.$$

Consequently,

$$
q - \tilde{Y} = \begin{cases} q - 1, & \tilde{Y} = 1, \\ q, & \tilde{Y} = 0. \end{cases}
$$

and the conditional second moment of the noisy gradient is

$$
\begin{aligned}
\mathbb{E}\big[\|\tilde{g}(\phi, \tilde{Y}; w^*)\|^2 \mid \phi\big] &= \mathbb{E}\big[(q - \tilde{Y})^2 \mid \phi\big]\,\|\phi\|^2 \\
&= \big(p_1(q - 1)^2 + p_0 q^2\big)\,\|\phi\|^2.
\end{aligned}
$$

Substituting $p_1 = q(1 - 2\eta) + \eta$ and $p_0 = 1 - p_1$ and expanding gives

$$p_1(q - 1)^2 + p_0 q^2 = (1 - 4\eta)\,q(1 - q) + \eta.$$

Therefore

$$\mathbb{E}\big[\|\tilde{g}(\phi, \tilde{Y}; w^*)\|^2 \mid \phi\big] = \Big((1 - 4\eta)\,q(1 - q) + \eta\Big)\,\|\phi\|^2.$$

Reinstating $q = q^*(\phi)$ yields

$$\mathbb{E}\big[\|\tilde{g}(\phi, \tilde{Y}; w^*)\|^2 \mid \phi\big] = \Big((1 - 4\eta)\,q^*(\phi)\big(1 - q^*(\phi)\big) + \eta\Big)\,\|\phi\|^2,$$

which is the form stated in Equation (6).

### F.4. Rollout-Level Mixture and Representation Norms

We formalize two facts used in Section 3.3: (i) rollout-level label mixture is an approximately unbiased proxy for the latent teacher mixture, and (ii) under bounded representation norms, average $q(1-q)$ and average $q(1-q)\|\phi\|^2$ differ only by constant factors.

**Label variance decomposition.** Fix a rollout $x$ with $n$ steps. For step $j$ let $q_j := q_{x,j}^* \in [0,1]$ and $Y_j \mid q_j \sim \text{Bernoulli}(q_j)$, independently conditioned on $\{q_j\}$. Define

$$\hat{p} := \frac{1}{n} \sum_{j=1}^n Y_j, \qquad \bar{q} := \frac{1}{n} \sum_{j=1}^n q_j, \qquad A(x) := \frac{1}{n} \sum_{j=1}^n q_j(1-q_j).$$

**Lemma 2** (Label variance decomposition). *Conditioned on $\{q_j\}$, the empirical label variance satisfies*

$$\mathbb{E}\big[\hat{p}(1-\hat{p}) \,\big|\, \{q_j\}\big] = \bar{q}(1-\bar{q}) - \frac{1}{n^2} \sum_{j=1}^n q_j(1-q_j).$$

*Proof.* We have

$$\hat{p} = \frac{1}{n} \sum_{j=1}^n Y_j, \qquad \hat{p}^2 = \frac{1}{n^2} \sum_{j=1}^n Y_j^2 + \frac{2}{n^2} \sum_{1 \le j < k \le n} Y_j Y_k.$$

Conditioned on $\{q_j\}$ the $Y_j$ are independent with $\mathbb{E}[Y_j \mid q_j] = q_j$ and $\mathbb{E}[Y_j^2 \mid q_j] = q_j$, so

$$\mathbb{E}[\hat{p} \mid \{q_j\}] = \bar{q}, \qquad \mathbb{E}[\hat{p}^2 \mid \{q_j\}] = \frac{1}{n^2} \sum_{j=1}^n q_j + \frac{2}{n^2} \sum_{1 \le j < k \le n} q_j q_k.$$

Using

$$\bar{q}^2 = \left( \frac{1}{n} \sum_{j=1}^n q_j \right)^2 = \frac{1}{n^2} \sum_{j=1}^n q_j^2 + \frac{2}{n^2} \sum_{1 \le j < k \le n} q_j q_k$$

to eliminate the cross terms yields

$$\mathbb{E}[\hat{p}^2 \mid \{q_j\}] = \bar{q}^2 + \frac{1}{n^2} \sum_{j=1}^n q_j - \frac{1}{n^2} \sum_{j=1}^n q_j^2 = \bar{q}^2 + \frac{1}{n^2} \sum_{j=1}^n q_j(1-q_j).$$

Finally,

$$\mathbb{E}\big[\hat{p}(1-\hat{p}) \,\big|\, \{q_j\}\big] = \mathbb{E}[\hat{p} \mid \{q_j\}] - \mathbb{E}[\hat{p}^2 \mid \{q_j\}] = \bar{q}(1-\bar{q}) - \frac{1}{n^2} \sum_{j=1}^n q_j(1-q_j),$$

as claimed. □

Since $t \mapsto t(1-t)$ is concave on $[0,1]$, Jensen's inequality gives

$$A(x) = \frac{1}{n} \sum_{j=1}^n q_j(1-q_j) \ \le \ \bar{q}(1-\bar{q}) =: \theta_x.$$

Using Lemma 2 and the identity

$$\frac{1}{n^2} \sum_{j=1}^n q_j(1-q_j) = \frac{1}{n} A(x),$$

we can rewrite

$$\mathbb{E}\big[\hat{p}(1-\hat{p}) \,\big|\, \{q_j\}\big] = \theta_x - \frac{1}{n} A(x).$$

Since $0 \le A(x) \le \theta_x$, this immediately yields the sandwich bound

$$\theta_x\left(1 - \frac{1}{n}\right) \;\le\; \mathbb{E}\big[\hat{p}(1 - \hat{p}) \,\big|\, \{q_j\}\big] \;\le\; \theta_x.$$

Thus the conditional bias of $\hat{p}(1 - \hat{p})$ as an estimator of the teacher-level mixture $\theta_x$ is at most $\theta_x/n \le 1/(4n)$, and $\hat{p}(1 - \hat{p})$ is an approximately unbiased proxy for $\theta_x$. In particular, rollouts with larger $\hat{p}(1 - \hat{p})$ tend to have larger teacher-level mixture $\theta_x$ (in expectation, up to an $\mathcal{O}(1/n)$ bias). Since $A(x) \le \theta_x$, a larger $\theta_x$ simply provides more headroom for $A(x)$ to be large, and therefore for the rollout to contain more informative steps.

**Symmetric flip noise and induced mixture.** In Section 3.3 we also consider a symmetric flip noise approximation: the observed label $\tilde{Y}_j$ is obtained by independently flipping the clean label $Y_j$ with probability $\eta \in [0, 1/2]$. Let $B_j \sim$ Bernoulli($\eta$) be independent of $Y_j$ and define $\tilde{Y}_j := Y_j \oplus B_j$. Conditioned on $q_j$, we have

$$\begin{aligned}
\tilde{q}_j \;:=\; \mathbb{P}(\tilde{Y}_j = 1 \mid q_j) &= \mathbb{P}(Y_j = 1, B_j = 0 \mid q_j) + \mathbb{P}(Y_j = 0, B_j = 1 \mid q_j) \\
&= (1 - \eta)q_j + \eta(1 - q_j) = (1 - 2\eta)q_j + \eta. \tag{22}
\end{aligned}$$

Thus $\tilde{Y}_j \mid q_j \sim$ Bernoulli($\tilde{q}_j$). Averaging (22) over steps gives

$$\bar{\tilde{q}} = \frac{1}{n}\sum_{j=1}^{n} \tilde{q}_j = (1 - 2\eta)\bar{q} + \eta.$$

Defining $\tilde{\theta}_x := \bar{\tilde{q}}(1 - \bar{\tilde{q}})$ and $\theta_x := \bar{q}(1 - \bar{q})$, a direct expansion yields

$$\begin{aligned}
\tilde{\theta}_x &= \big((1 - 2\eta)\bar{q} + \eta\big)\Big(1 - (1 - 2\eta)\bar{q} - \eta\Big) \\
&= \big((1 - 2\eta)\bar{q} + \eta\big)\big((1 - \eta) - (1 - 2\eta)\bar{q}\big) \\
&= (1 - 2\eta)^2\,\bar{q}(1 - \bar{q}) + \eta(1 - \eta) \\
&= (1 - 2\eta)^2\,\theta_x + \eta(1 - \eta). \tag{23}
\end{aligned}$$

**Closeness to the noise-free analysis for small $\eta$.** Eq. (23) implies

$$\tilde{\theta}_x - \theta_x = \big((1 - 2\eta)^2 - 1\big)\theta_x + \eta(1 - \eta),$$

so using $0 \le \theta_x \le 1/4$ we obtain the uniform bound

$$|\tilde{\theta}_x - \theta_x| \le 4\eta\,\theta_x + \eta(1 - \eta) \le 2\eta. \tag{24}$$

Moreover, since $\tilde{Y}_{x,j} \mid \tilde{q}_{x,j} \sim$ Bernoulli($\tilde{q}_{x,j}$) independently conditioned on $\{\tilde{q}_{x,j}\}$, applying Lemma 2 with $q_{x,j}$ replaced by $\tilde{q}_{x,j}$ shows that $\hat{p}_{\sim,x}(1 - \hat{p}_{\sim,x})$ estimates $\tilde{\theta}_x$ up to an additional $\mathcal{O}(1/n)$ bias. Therefore, when $\eta$ is small (and $n$ is not too small), the mixture computed from observed labels is within $\mathcal{O}(\eta) + \mathcal{O}(1/n)$ of the noise-free target in expectation, so the analysis in the noise-free setting applies up to a small additive perturbation.

**Effect of bounded representation norms.** Define the full average step-wise information

$$A_{\text{full}}(x) := \frac{1}{n}\sum_{j=1}^{n} q_j(1 - q_j)\,\|\phi_{x,j}\|^2.$$

Assume the representations are uniformly bounded: there exist constants $0 < c_{\min} \le c_{\max} < \infty$ such that $c_{\min} \le \|\phi_{x,j}\|^2 \le c_{\max}$ for all steps. Then

$$c_{\min} A(x) \;\le\; A_{\text{full}}(x) \;\le\; c_{\max} A(x).$$

Hence, up to global multiplicative constants, $A(x)$ and $A_{\text{full}}(x)$ measure the same notion of step-wise information. Qualitative comparisons between rollouts can therefore be phrased in terms of $A(x)$. In particular, increasing $A(x)$ increases a corresponding lower bound on $A_{\text{full}}(x)$, and $A(x)$ serves as a constant-factor proxy for $A_{\text{full}}(x)$ under this boundedness assumption.

### F.5. MC-induced Pseudo-positive Probability and Monotonicity

**Posterior and closed form.** Let $r \in [0,1]$ denote the one-shot success probability of a step, and let $K \mid r \sim \text{Binomial}(N, r)$ be the number of successful continuations. Fix $\tau \in (0,1)$ and define $\tau$-reliability by $Z := \mathbb{I}[r \geq \tau]$. We place a Beta prior $r \sim \text{Beta}(a, b)$ with density $p(r) \propto r^{a-1}(1-r)^{b-1}$ for $a, b > 0$. The Binomial likelihood is

$$p(K = k \mid r) = \binom{N}{k} r^k (1-r)^{N-k}.$$

By Bayes' rule,

$$p(r \mid K = k) \propto p(K = k \mid r) \, p(r) \propto r^{a+k-1}(1-r)^{b+N-k-1},$$

which is the density of $\text{Beta}(a + k, \ b + N - k)$. Writing $\alpha_k := a + k$ and $\beta_k := b + N - k$, the normalized posterior density is

$$f_k(r) := p(r \mid K = k) = \frac{1}{B(\alpha_k, \beta_k)} r^{\alpha_k - 1}(1-r)^{\beta_k - 1}, \qquad r \in (0,1),$$

where $B(\alpha, \beta) = \int_0^1 t^{\alpha-1}(1-t)^{\beta-1} \, dt$ is the Beta function. Define the conditional pseudo-positive probability (effective noise level)

$$\eta_{\text{eff}}(k) := \Pr(Z = 0 \mid K = k) = \Pr(r < \tau \mid K = k) = \int_0^\tau f_k(r) \, dr.$$

Let $B(\tau; \alpha, \beta) := \int_0^\tau t^{\alpha-1}(1-t)^{\beta-1} \, dt$ be the incomplete Beta function. Then

$$\eta_{\text{eff}}(k) = \frac{B(\tau; \alpha_k, \beta_k)}{B(\alpha_k, \beta_k)} =: I_\tau(\alpha_k, \beta_k),$$

where $I_\tau(\alpha, \beta)$ is the regularized incomplete beta function.

**Monotonicity.**

**Lemma 3** (Monotonicity of $\eta_{\text{eff}}$). *For any $a, b > 0$, $N \geq 1$, and $\tau \in (0,1)$, the map $k \mapsto \eta_{\text{eff}}(k) = \Pr(r < \tau \mid K = k)$ is strictly decreasing on $\{0, 1, \ldots, N\}$. Equivalently, $\Pr(r \geq \tau \mid K = k) = 1 - \eta_{\text{eff}}(k)$ is strictly increasing in $k$.*

*Proof.* Let $f_k$ be the posterior density above with parameters $\alpha_k = a + k$ and $\beta_k = b + N - k$, i.e.,

$$f_k(r) = \frac{1}{B(\alpha_k, \beta_k)} r^{\alpha_k - 1}(1-r)^{\beta_k - 1}, \qquad r \in (0,1).$$

Fix $k \in \{0, \ldots, N-1\}$ and compare consecutive posteriors. A direct calculation gives, for $r \in (0,1)$,

$$\frac{f_{k+1}(r)}{f_k(r)} = \frac{B(\alpha_k, \beta_k)}{B(\alpha_{k+1}, \beta_{k+1})} \cdot r^{\alpha_{k+1} - \alpha_k}(1-r)^{\beta_{k+1} - \beta_k}$$

$$= \frac{B(\alpha_k, \beta_k)}{B(\alpha_k + 1, \beta_k - 1)} \cdot \frac{r}{1-r}.$$

Since $k \leq N - 1$ and $b > 0$, we have $\beta_k = b + N - k \geq b + 1 > 1$, so $\beta_k - 1 > 0$ and the Beta-function identity $B(\alpha + 1, \beta - 1) = \frac{\alpha}{\beta - 1} B(\alpha, \beta)$ applies. Thus we obtain the explicit form

$$\frac{f_{k+1}(r)}{f_k(r)} = C_k \frac{r}{1-r}, \qquad C_k := \frac{\beta_k - 1}{\alpha_k} = \frac{b + N - k - 1}{a + k},$$

which is strictly increasing in $r$ on $(0,1)$ since $r/(1-r)$ is strictly increasing.

Fix $\tau \in (0,1)$ and define $c := \frac{f_{k+1}(\tau)}{f_k(\tau)} = C_k \frac{\tau}{1-\tau}$. Because $\frac{f_{k+1}(r)}{f_k(r)}$ is strictly increasing in $r$, we have

$$\frac{f_{k+1}(r)}{f_k(r)} < c \quad \text{for } r \in (0, \tau), \qquad \frac{f_{k+1}(r)}{f_k(r)} > c \quad \text{for } r \in (\tau, 1),$$

and $\frac{f_{k+1}(\tau)}{f_k(\tau)} = c$ at $r = \tau$. Multiplying by $f_k(r)$ and integrating yields

$$\int_0^\tau f_{k+1}(r)\,dr \leq c \int_0^\tau f_k(r)\,dr, \qquad \int_\tau^1 f_{k+1}(r)\,dr \geq c \int_\tau^1 f_k(r)\,dr.$$

Let $A_k := \int_0^\tau f_k(r)\,dr = \Pr(r < \tau \mid K = k)$. Since $\int_0^1 f_k = 1$, the second inequality becomes $1 - A_{k+1} \geq c(1 - A_k)$.

If $c \leq 1$, then the first inequality gives $A_{k+1} \leq cA_k \leq A_k$. If $c \geq 1$, then the second inequality implies $1 - A_{k+1} \geq 1 - A_k$, i.e. $A_{k+1} \leq A_k$. Thus in all cases $A_{k+1} \leq A_k$, proving monotonic non-increase.

Finally, since the ratio $\frac{f_{k+1}(r)}{f_k(r)}$ is *strictly* increasing in $r$, the inequalities $\frac{f_{k+1}(r)}{f_k(r)} < c$ on $(0, \tau)$ and $\frac{f_{k+1}(r)}{f_k(r)} > c$ on $(\tau, 1)$ are strict on sets of positive Lebesgue measure. Moreover, $f_k(r) > 0$ for all $r \in (0, 1)$ when $\alpha_k, \beta_k > 0$. Hence the integral inequalities above are strict, yielding $A_{k+1} < A_k$ for any $\tau \in (0, 1)$. Therefore $\eta_{\mathrm{eff}}(k) = A_k$ is strictly decreasing in $k$ on $\{0, 1, \ldots, N\}$. $\qquad\square$

# G. Training Dynamics

Figure 7,8 and 9 report the training dynamics of BIS-$\rho$ and Random-$\rho$ under keep ratios $\rho \in \{5, 10, 15, 25, 35, 50\}\%$. For each $\rho$, we track both the overall micro-F1 and the per-source macro-F1 on VisualProcessBench over training steps, and include the full-data reference as horizontal dashed lines. Across ratios and backbones, BIS not only achieves stronger final performance, but also improves faster: it reaches high accuracy in substantially fewer steps and maintains clear advantages over random sub-sampling throughout training. This gap is most pronounced in the low-budget regime, where Random-$\rho$ often learns slowly and remains far below the full-data reference, while BIS-$\rho$ rapidly closes the gap and frequently approaches full-data performance early in training.

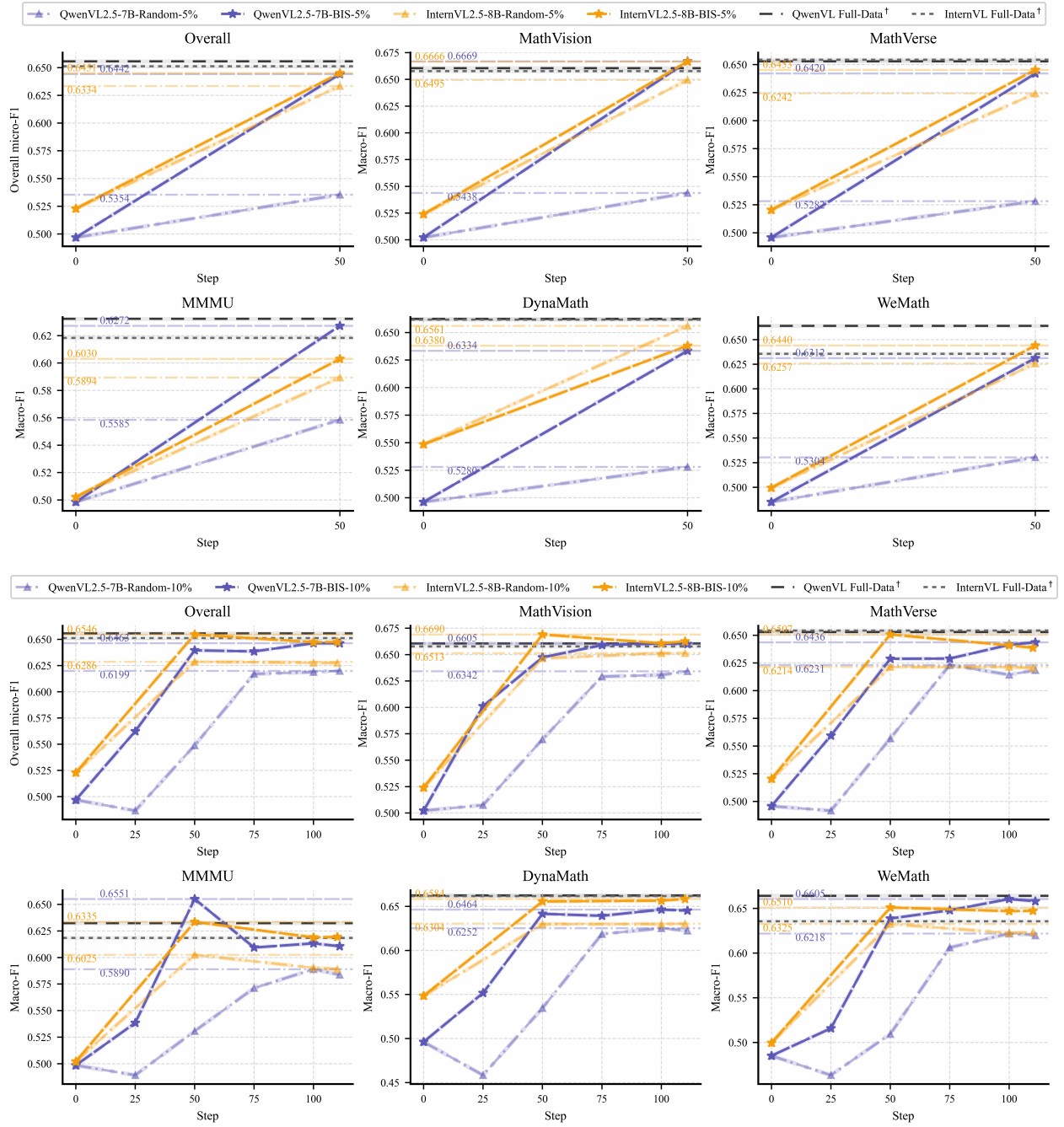

*Figure 7.* Training dynamics on VisualProcessBench for $\rho \in \{5\%, 10\%\}$, comparing BIS-$\rho$ and Random-$\rho$ for both InternVL2.5-8B and Qwen2.5-VL-7B (overall micro-F1 and per-source macro-F1).

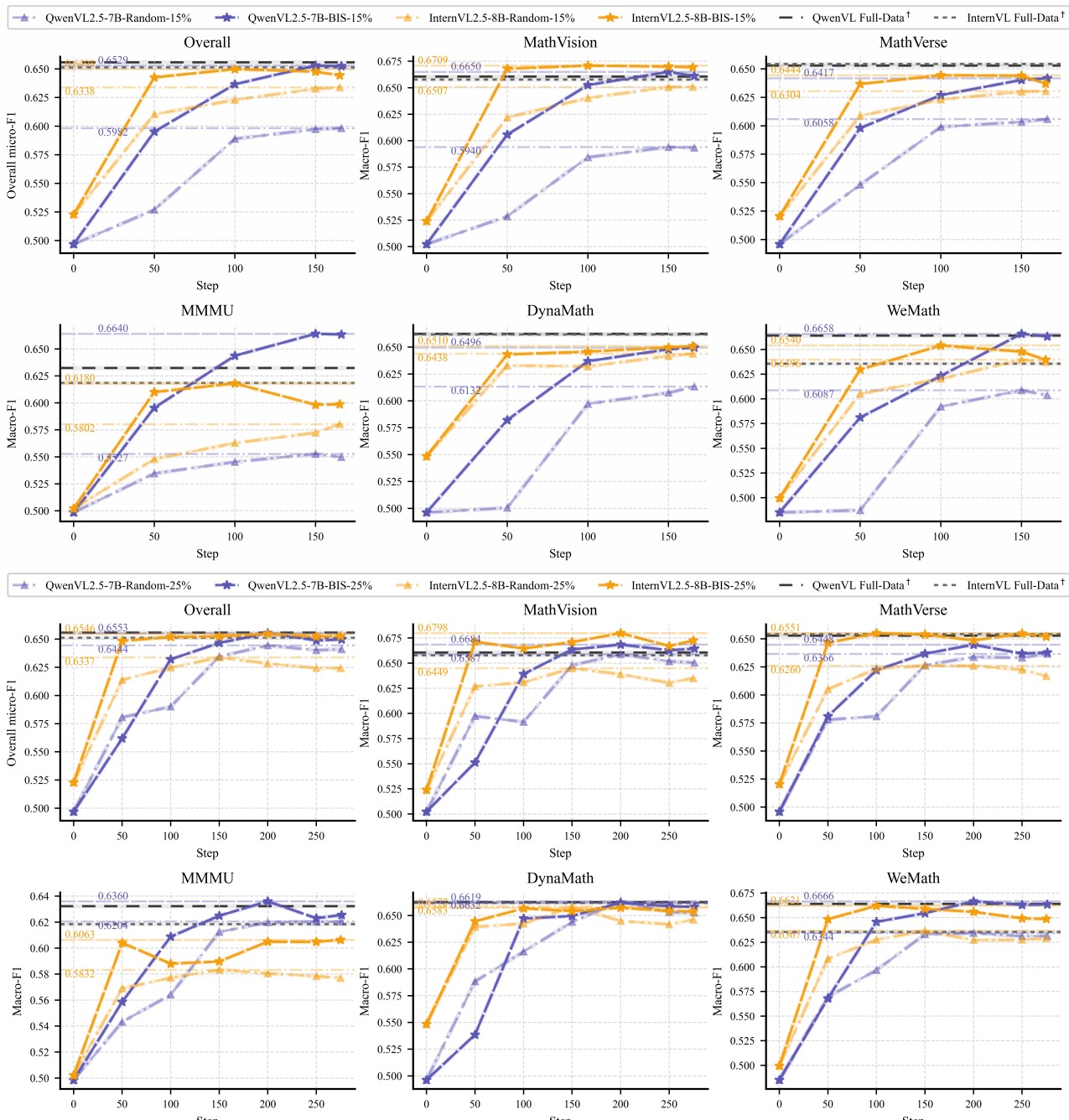

*Figure 8.* Training dynamics on VisualProcessBench for $\rho \in \{15\%, 25\%\}$, comparing BIS-$\rho$ and Random-$\rho$ for both InternVL2.5-8B and Qwen2.5-VL-7B (overall micro-F1 and per-source macro-F1).

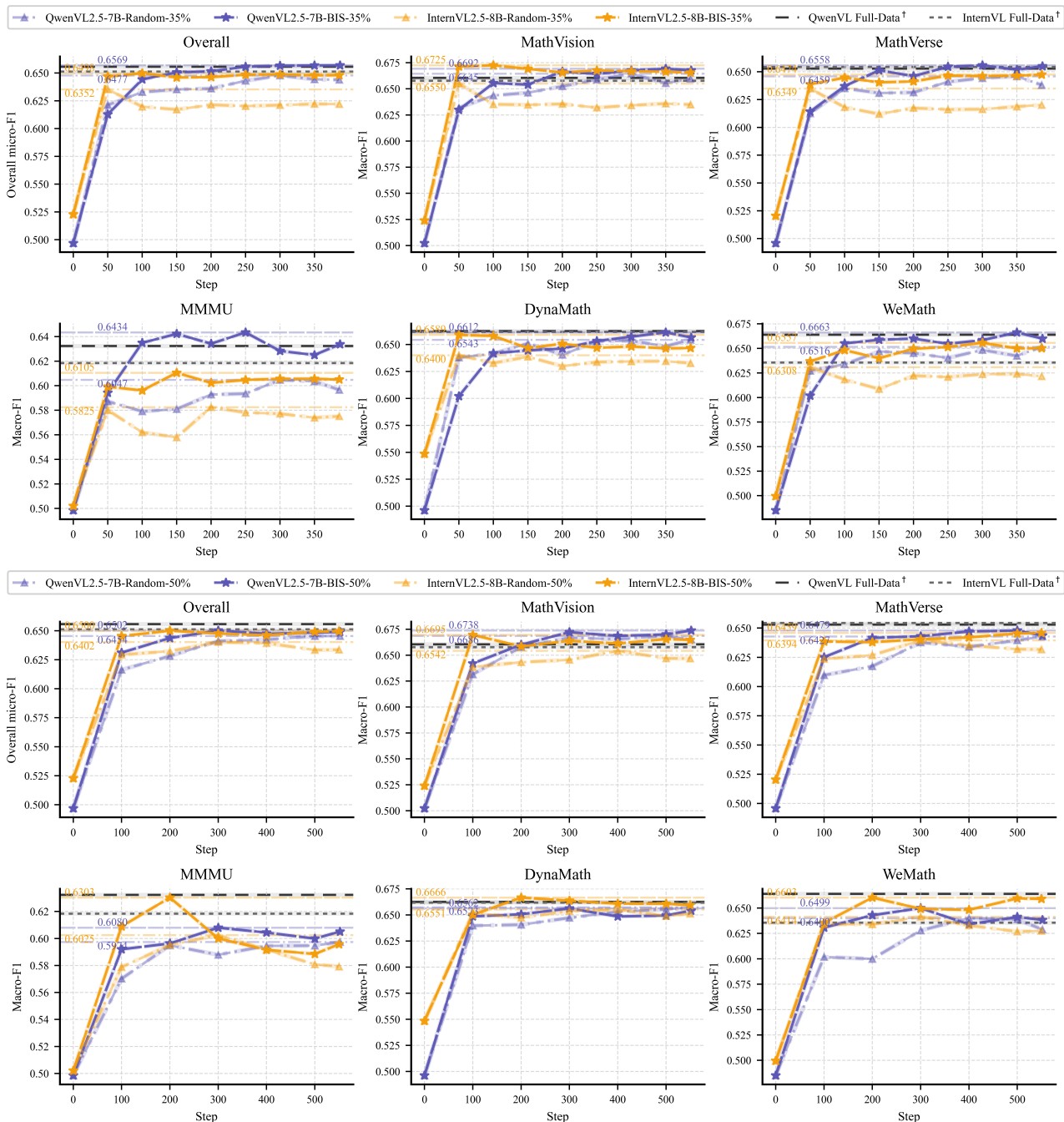

*Figure 9.* Training dynamics on VisualProcessBench for $\rho \in \{35\%, 50\%\}$, comparing BIS-$\rho$ and Random-$\rho$ for both InternVL2.5-8B and Qwen2.5-VL-7B (overall micro-F1 and per-source macro-F1).

## H. Per-source BIS Histograms

**Interpreting low-BIS sources via step-level MC difficulty.** Figure 10 reports, for each of the 38 source subsets, the BIS score distribution over all rollouts and the per-source top-25% rollouts selected by BIS. The per-source shapes vary substantially, and interpreting them requires care: because $\text{BIS}(x) = (p_{\text{pos}}(x)(1 - p_{\text{pos}}(x)) + \alpha)\, R(x)$, rollouts that are nearly pure (almost all-positive or almost all-negative) receive small mixture weight $p_{\text{pos}}(1 - p_{\text{pos}})$ and thus tend to concentrate at the lower end of the BIS range, even though these two purity regimes correspond to opposite difficulty profiles.

To disambiguate them, we pair the histograms with the step-level average MC score $\bar{s}$ for each source (Table 8), where $\bar{s}$ averages MC scores over all annotated steps and serves as a proxy for how often the MC solver can reach a correct completion from intermediate states. Under this lens, sources whose full-pool histogram is concentrated near low BIS scores split into two qualitatively different categories:

**High-$\bar{s}$ sources (easy for the current MC solver).** When $\bar{s}$ is high, low BIS scores mainly reflect trajectories that are dominated by positive steps ($p_{\text{pos}} \approx 1$), hence little label mixture. This is exemplified by *ScienceQA* ($\bar{s} = 0.9723$) and *NLVR2* ($\bar{s} = 0.9672$), as well as other high-$\bar{s}$ sources such as *DVQA* (0.9453). In these sources, naturally mixed trajectories are rarer, so the available high-BIS tail is intrinsically thinner.

**Low-$\bar{s}$ sources (hard for the current MC solver).** When $\bar{s}$ is low, low BIS scores instead reflect trajectories with few positive steps under MC sampling ($p_{\text{pos}} \approx 0$). Representative examples include *GeoQA+ (Open)* ($\bar{s} = 0.5720$) and *GeomVerse* (0.6819). Here the bottleneck is the scarcity of reliable positive anchors: even when errors are plentiful, many steps rarely lead to successful continuations, so the pool contains fewer rollouts that simultaneously provide informative negatives and trustworthy positives.

Between these extremes, **medium-$\bar{s}$ sources** tend to provide the richest substrate for BIS, because they naturally generate rollouts with non-trivial label mixture while still retaining a meaningful fraction of reasonably reliable positives. For example, *FigureQA* ($\bar{s} = 0.7615$) and *VQAv2* ($\bar{s} = 0.8253$) exhibit visibly heavier mass at moderate-to-high BIS scores, indicating a larger supply of mixed-but-reliable trajectories that BIS can exploit.

Across all sources, BIS selection produces a consistent within-source reallocation: the orange histograms shift weight toward higher scores relative to the blue ones, meaning that under the same per-source budget (top-25% per file) BIS preferentially keeps the most mixed and reliable rollouts available in that source. This provides a distributional explanation for the main experimental trend in Table 2: compared with Random-25%, which spends budget proportionally to the original pool, and Low-MC-25%, which can over-focus on unreliable pseudo-positives, BIS-25% systematically extracts the highest-information tail when such a tail exists (e.g., medium-$\bar{s}$ sources), while avoiding wasting budget on uninformative near-pure trajectories in very easy sources or on noise-dominated trajectories in very hard sources. As a result, BIS allocates the fixed update budget to a subset with higher effective signal per update, aligning with its consistent gains over Random / Mixed / Low-MC under the same rollout and optimization budgets.

**Consistent selection patterns in mixture and reliability.** Figure 11 and 12 further reports, for each source, the distributions of the reliability term $R(x)$ and the mixture term $p_{\text{pos}}(x)(1 - p_{\text{pos}}(x))$ for all rollouts and the selected top-25% subset, together with the coverage curve (Selected / All). Across sources, BIS follows two stable behaviors: it strongly suppresses low-$R(x)$ rollouts, and it consistently favors higher mixture by shifting coverage toward moderate-to-high $p_{\text{pos}}(1 - p_{\text{pos}})$. Importantly, the preference over $R(x)$ is not monotonic: coverage often peaks at moderate $R(x)$ and then saturates or drops as $R(x) \to 1$, since near-pure trajectories can have large $R(x)$ but still receive small mixture weight and thus low BIS scores. The plots also highlight substantial source-level heterogeneity in the available mixed-and-reliable tail, complementing the step-level MC difficulty analysis in Table 8.

*Table 8.* Per-source step-level MC score statistics on VisualPRM400K-v1.1. Avg. MC is the mean MC score over all annotated steps; Avg./16 equals $16\times$Avg. MC (expected successes out of $N=16$ continuations). Step/Rollout counts follow the same validity filtering as in our histogram analysis.

| Subset | Avg. MC | Avg./16 | # Steps | # Rollouts |
|---|---|---|---|---|
| AI2D (Kembhavi et al., 2016) | 0.9409 | 15.0539 | 146,419 | 28,147 |
| ChartQA (Masry et al., 2022) | 0.9078 | 14.5245 | 136,655 | 28,049 |
| CLEVR (Johnson et al., 2017) | 0.9106 | 14.5689 | 109,317 | 24,004 |
| COCO-ReM (Y/N) (Singh et al., 2024) | 0.8933 | 14.2922 | 55,628 | 18,450 |
| DocVQA (Mathew et al., 2021) | 0.8663 | 13.8604 | 85,589 | 21,049 |
| DVQA (Kafle et al., 2018) | 0.9453 | 15.1255 | 177,352 | 35,367 |
| FigureQA (Kahou et al., 2017) | 0.7615 | 12.1843 | 57,601 | 12,345 |
| Geo170K (Gao et al., 2025) | 0.9023 | 14.4373 | 61,314 | 8,205 |
| Geometry3K (Lu et al., 2021a) | 0.9243 | 14.7880 | 94,624 | 11,756 |
| Geometry3K (Open) (Lu et al., 2021a) | 0.8707 | 13.9318 | 58,569 | 7,090 |
| GeomVerse (Kazemi et al., 2024) | 0.6819 | 10.9106 | 38,156 | 7,765 |
| GeoQA+ (Open) (Cao & Xiao, 2022) | 0.5720 | 9.1514 | 45,287 | 7,025 |
| GeoQA+ (MC) (Cao & Xiao, 2022) | 0.8486 | 13.5772 | 67,715 | 9,216 |
| GEOS (Seo et al., 2015) | 0.9061 | 14.4970 | 16,847 | 1,827 |
| GEOS (Open) (Seo et al., 2015) | 0.8217 | 13.1478 | 1,563 | 178 |
| IconQA (Lu et al., 2021b) | 0.8908 | 14.2536 | 121,873 | 25,811 |
| InfographicVQA (Mathew et al., 2022) | 0.7645 | 12.2323 | 78,408 | 17,996 |
| KonIQ-10k (Hosu et al., 2020) | 0.8822 | 14.1148 | 72,430 | 13,337 |
| M3CoT (Chen et al., 2024b) | 0.8811 | 14.0971 | 115,649 | 19,476 |
| MAPQA-SUV (Chang et al., 2022) | 0.7260 | 11.6155 | 48,368 | 12,366 |
| MathV360K (Shi et al., 2024) | 0.8759 | 14.0142 | 107,503 | 20,132 |
| MAVIS-Abs (Zhang et al., 2025c) | 0.8104 | 12.9670 | 166,756 | 16,530 |
| MAVIS-Cos (Zhang et al., 2025c) | 0.8057 | 12.8916 | 86,393 | 9,174 |
| MAVIS-Log (Zhang et al., 2025c) | 0.7357 | 11.7719 | 162,713 | 15,422 |
| MAVIS-Poly (Zhang et al., 2025c) | 0.8836 | 14.1375 | 206,293 | 23,461 |
| MAVIS-Sin (Zhang et al., 2025c) | 0.8231 | 13.1701 | 88,130 | 9,675 |
| MAVIS-Tan (Zhang et al., 2025c) | 0.8027 | 12.8428 | 60,391 | 6,180 |
| MAVIS-Geo-0 (Zhang et al., 2025c) | 0.9280 | 14.8473 | 78,764 | 13,149 |
| MAVIS-Geo-1 (Zhang et al., 2025c) | 0.8765 | 14.0243 | 37,760 | 6,728 |
| MAVIS-Geo-2 (Zhang et al., 2025c) | 0.7990 | 12.7848 | 25,851 | 4,801 |
| MAVIS-Geo-3 (Zhang et al., 2025c) | 0.7719 | 12.3508 | 19,080 | 3,706 |
| NLVR2 (Suhr et al., 2019) | 0.9672 | 15.4753 | 96,072 | 26,439 |
| ScienceQA (Lu et al., 2022) | 0.9723 | 15.5574 | 186,894 | 34,523 |
| SROIE (Huang et al., 2019) | 0.8708 | 13.9325 | 83,389 | 18,074 |
| Super-CLEVR (Int) (Li et al., 2023) | 0.7325 | 11.7205 | 37,279 | 12,070 |
| Super-CLEVR (Y/N) (Li et al., 2023) | 0.7713 | 12.3404 | 33,171 | 10,440 |
| UniGeo (Chen et al., 2022) | 0.6015 | 9.6239 | 41,523 | 6,299 |
| VQAv2 (Goyal et al., 2017) | 0.8253 | 13.2043 | 67,166 | 18,887 |

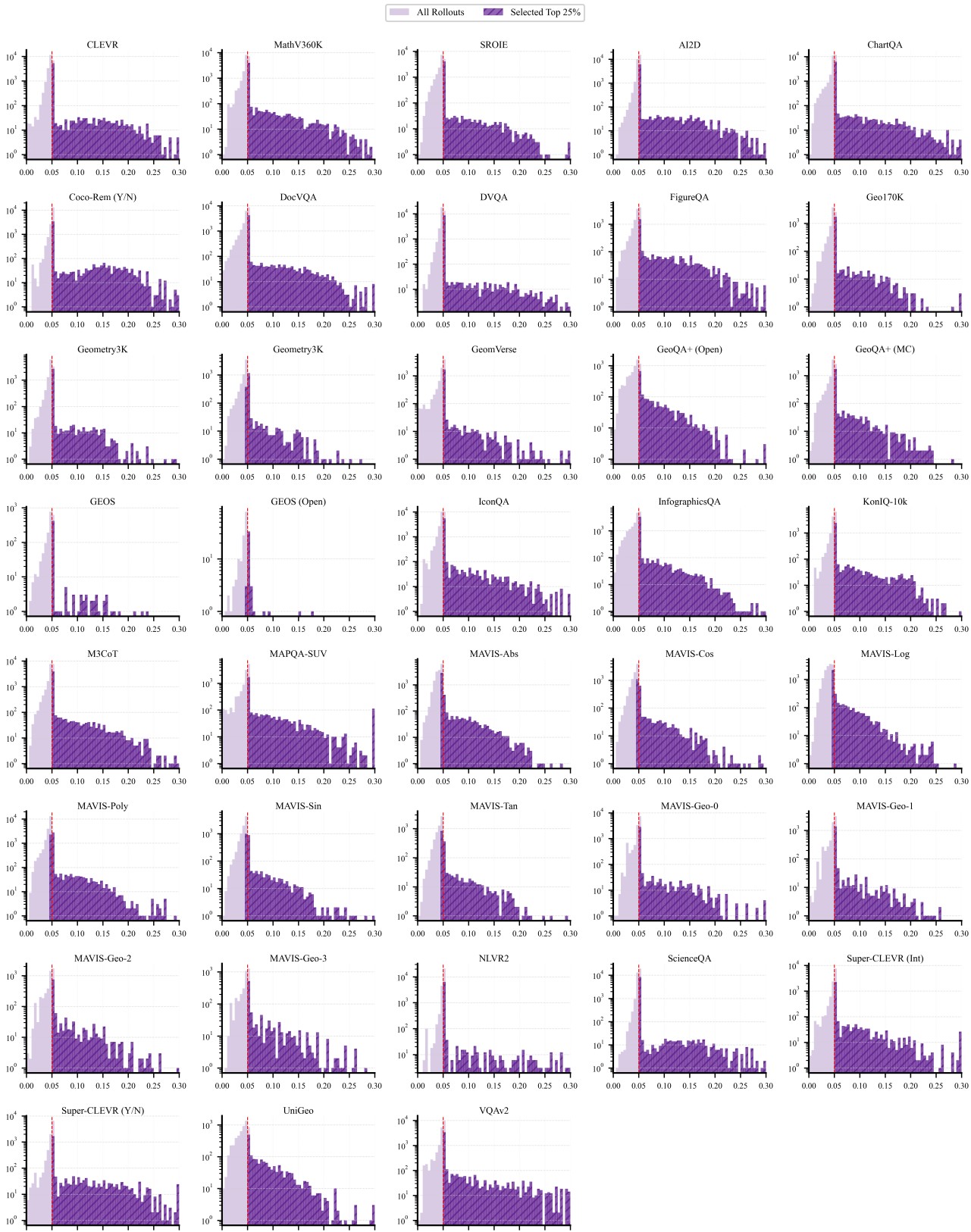

*Figure 10.* Per-source BIS score distributions on VisualPRM400K-v1.1. Each panel corresponds to one source dataset; light purple bars show all rollouts from that source and dark purple bars show the per-file top-25% rollouts selected by BIS. Across sources, the full-pool distributions differ markedly—with some datasets exhibiting sharp spikes near the baseline $\alpha = 0.05$ and others showing heavier high-score tails—but BIS consistently suppresses low-score mass and enriches higher-score regions within each source.

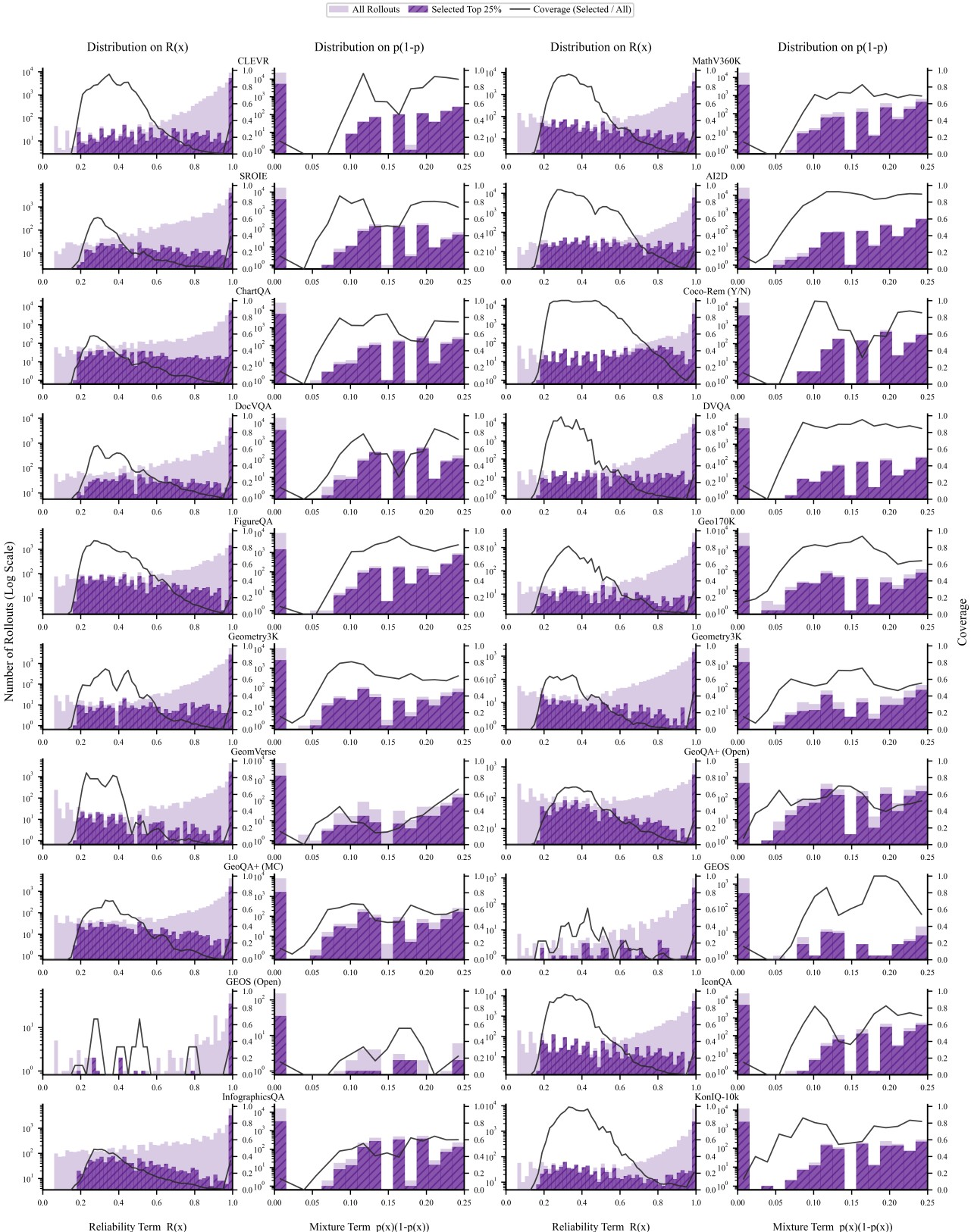

*Figure 11.* Per-source component histograms on VisualPRM400K-v1.1 for the reliability term $R(x)$ and mixture term $p_{\mathrm{pos}}(x)(1-p_{\mathrm{pos}}(x))$, comparing all rollouts and BIS top-25%. Black curve: coverage (Selected / All).

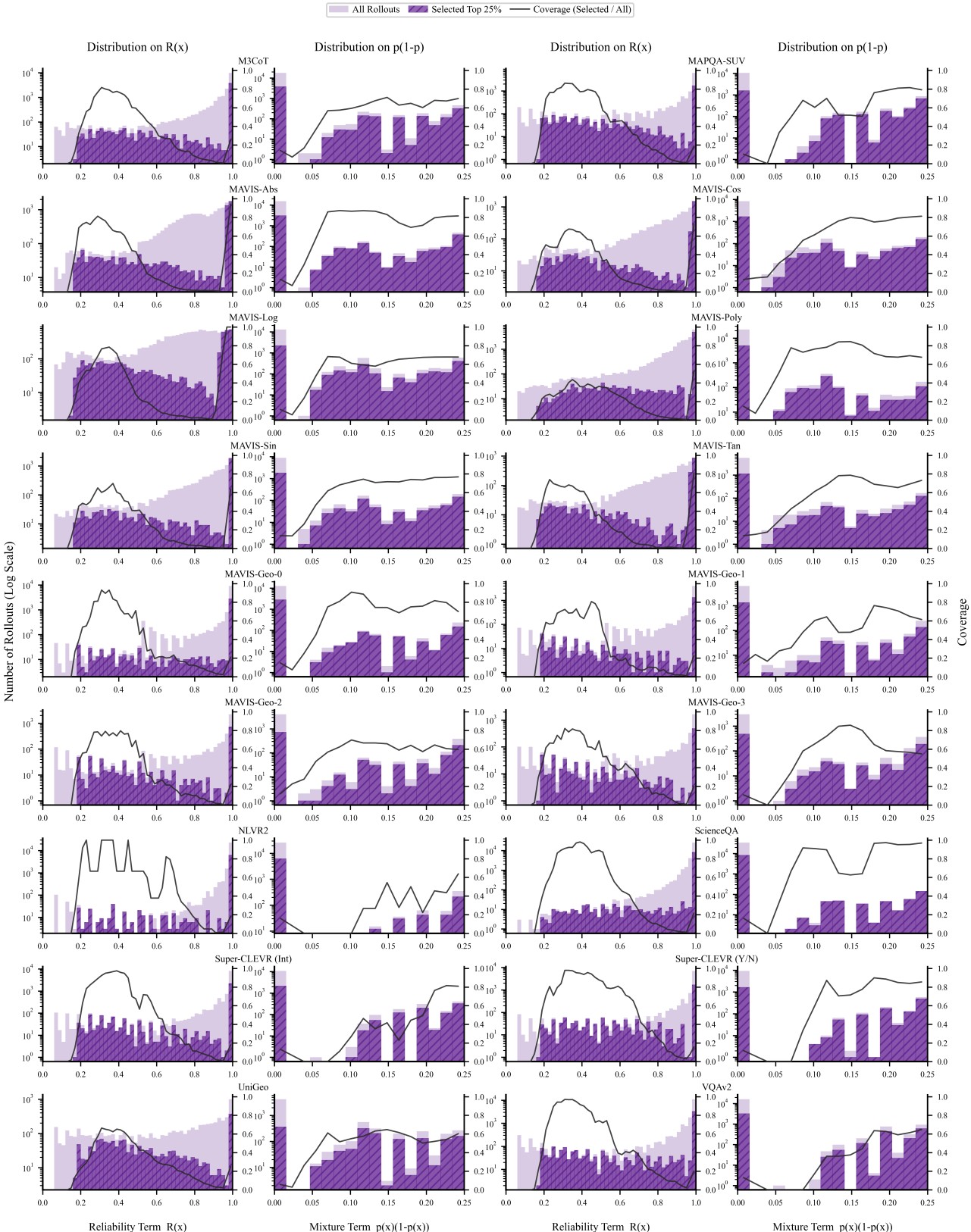

*Figure 12.* Per-source component histograms on VisualPRM400K-v1.1 for the reliability term $R(x)$ and mixture term $p_{pos}(x)(1-p_{pos}(x))$, comparing all rollouts and BIS top-25%. Black curve: coverage (Selected / All).

**Case Study 1**

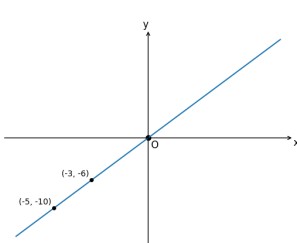

**Question.**
The graph of the function $f(x) = a * x + b$ is shown, where the condition that $a$ is non-zero is satisfied. The function $f(x)$ takes the values $-10$ and $-6$ at $x = -5$ and $x = -3$, respectively. Given the graph and the aforementioned conditions, identify the zeros of the function.

**Ground truth.** $0$

**Step scores** *(hard label $y_j = \mathbb{I}[score_j > 0]$).*

| Step | Content | Score | $y_j$ |
|------|---------|-------|-------|
| Step-0 | To determine the zeros of the function $f(x) = ax + b$, we need to follow these steps: | 0.875 | 1 |
| Step-1 | 1. **Identify the equation of the line:** 
 - Given that $f(x) = -10$ at $x = -5$ and $f(x) = -6$ at $x = -3$, we can use these points to find the slope $a$ and the y-intercept $b$. 
 - Substitute $x = -5$, $y = -10$ into the equation: $-10 = a(-5) + b$   (1) 
 - Substitute $x = -3$, $y = -6$ into the equation: $-6 = a(-3) + b$   (2) | 0.875 | 1 |
| Step-2 | 2. **Solve the system of equations:** 
 - From equation (1): $-10 = -5a + b$ 
 - From equation (2): $-6 = -3a + b$ | 0.9375 | 1 |
| Step-3 | 3. **Simplify the equations:** 
 - Equation (1) becomes: $b = -10 + 5a$ 
 - Equation (2) becomes: $-6 = -3a + b$ | 0.875 | 1 |
| Step-4 | 4. **Substitute $b$ values from equation (1) into equation (2):** 
 - Substitute $b = -10 + 5a$ into $-6 = -3a + b$: $-6 = -3a + (-10 + 5a)$ 
 - Simplify: $-6 = -3a - 10 + 5a$ 
 - Combine like terms: $-6 = 2a - 10$ 
 - Solve for $a$: $6 = 2a \implies a = 3$ | 0.0 | 0 |
| Step-5 | 5. **Find $b$ using the value of $a$:** 
 - Substitute $a = 3$ back into $b = -10 + 5a$: $b = -10 + 5(3) = -10 + 15 = 5$ | 0.0 | 0 |
| Step-6 | 6. **Write the final equation of the function:** 
 - The equation of the function is $f(x) = 3x + 5$. | 0.0 | 0 |
| Step-7 | 7. **Setting $f(x) = 0$ to find the zeros:** 
 - $0 = 3x + 5$ 
 - Solve for $x$: $3x = -5 \implies x = -\dfrac{5}{3}$ | 0.0 | 0 |
| Step-8 | Thus, the zeros of the function $f(x) = 3x + 5$ are $x = -\frac{5}{3}$. | 0.0 | 0 |
| Step-9 | Final answer: $-\frac{5}{3}$ | 0.0 | 0 |

**BIS computation** *($\alpha = 0.05$).*
Valid step scores are the 10 entries above. Hard labels: $y_j = 1$ iff $score_j > 0$.

$$n = 10, \qquad n_{\text{pos}} = 4, \qquad p_{\text{pos}} = \frac{4}{10} = 0.4,$$

$$R = \frac{0.875 + 0.875 + 0.9375 + 0.875}{4} = 0.890625,$$

$$\text{BIS}(x) = \big(p_{\text{pos}}(1 - p_{\text{pos}}) + 0.05\big) \cdot R$$

$$= (0.4 \cdot 0.6 + 0.05) \cdot 0.890625 = 0.25828125.$$

*Figure 13.* Case study (MAVIS-Function (Zhang et al., 2025c)): step-level MC scores, hard labels, and BIS computation for one rollout.

## Case Study 2

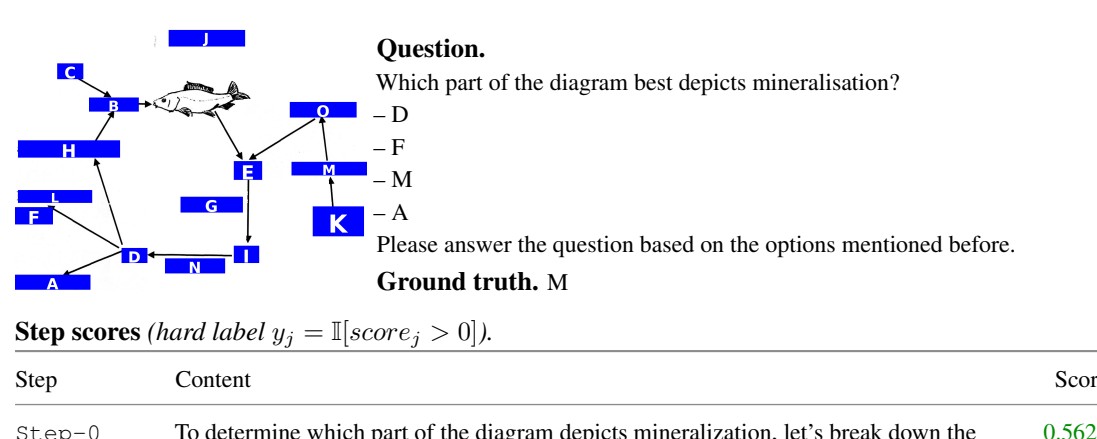

**Question.**
Which part of the diagram best depicts mineralisation?

– D
– F
– M
– A

Please answer the question based on the options mentioned before.

**Ground truth.** M

**Step scores** *(hard label $y_j = \mathbb{I}[score_j > 0]$).*

| Step | Content | Score | $y_j$ |
|------|---------|-------|-------|
| Step-0 | To determine which part of the diagram depicts mineralization, let's break down the relevant concepts. | 0.5625 | 1 |
| Step-1 | 1. **Mineralization**: It refers to the process by which organic matter is converted into minerals. This often involves bacteria breaking down organic materials and releasing inorganic salts such as nitrates and phosphates back into the soil, accessible for plant uptake. | 0.6875 | 1 |
| Step-2 | 2. **Flowchart Analysis**:
- Identify the main components in the diagram that are involved with mineralization.
- Typically, mineralization follows the decomposition pathway, so look for arrows leading to outputs that could be soil minerals. | 0.6875 | 1 |
| Step-3 | 3. **Options**:
- D: This point may involve decomposers like bacteria and fungi that convert organic matter to minerals.
- F: This appears to be involved in detritus or initial organic matter feed.
- M: This might be part of the nutrient cycle but may not directly depict mineralization.
- A: This point seems more related to primary producers or initial growth stage. | 0.0625 | 1 |
| Step-4 | 4. **Evaluation of Options**:
- D shows arrows indicating a transformation from organic to inorganic content, implying the breakdown and conversion into minerals.
- F, M, A do not visibly indicate this mineralization process. | 0.0 | 0 |
| Step-5 | ### Answer: | 0.0 | 0 |
| Step-6 | Based on these analyses, D is the most consistent signifier of the mineralization process in this diagram. Therefore, the answer is: | 0.0 | 0 |
| Step-7 | Final answer: D | 0.0 | 0 |

**BIS computation** *($\alpha = 0.05$).*
Valid step scores are the 8 entries above. Hard labels: $y_j = 1$ iff $score_j > 0$.

$$n = 8, \qquad n_{\text{pos}} = 4, \qquad p_{\text{pos}} = \frac{4}{8} = 0.5,$$

$$R = \frac{0.5625 + 0.6875 + 0.6875 + 0.0625}{4} = 0.5,$$

$$\text{BIS}(x) = \big(p_{\text{pos}}(1 - p_{\text{pos}}) + 0.05\big) \cdot R$$

$$= (0.5 \cdot 0.5 + 0.05) \cdot 0.5 = 0.15.$$

*Figure 14.* Case study (AI2D (Kembhavi et al., 2016)): step-level MC scores, hard labels, and BIS computation for one rollout.

**Case Study 3**

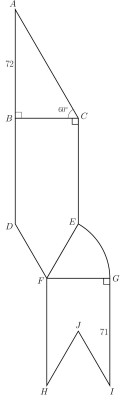

**Question.**

Angle ABC is equivalent to $\pi/2$. Side DE extending into an equilateral triangle shape beyond the rectangle. EFG forms a sector. Side HI continues into an equilateral triangle inside the rectangle. AB is equal to 72, Angle ACB measures $60°$, GI measures 71. What is the total surface area of the shape GFHJI?

A. 954*$\sqrt{3}$

B. 3816*$\sqrt{3}$/5

C. 70

D. 1272*$\sqrt{3}$

**Ground truth.** D

**Step scores** *(hard label $y_j = \mathbb{I}[\text{score}_j > 0]$).*

| Step | Content | Score | $y_j$ |
|------|---------|-------|-------|
| Step-1 | To find the total surface area of the complex shape GFHJI, let's break it down step by step. | 0.0625 | 1 |
| Step-2 | 1. **Identify the components of the shape GFHJI:** - Rectangle GFHI - Equilateral triangle HIJ - Segments GH and HI, which are parts of the equilateral triangle - The sector EFG | 0.0625 | 1 |
| Step-3 | 2. **Calculate the area of rectangle GFHI:** - Side GF = 71 (given) - Determine the width (HI) of the rectangle: - HI is part of the equilateral triangle HIJ. - The side of the equilateral triangle is GH, which is the same as CI (71 units). - The height of the equilateral triangle HIJ can be calculated as $\frac{\sqrt{3}}{2} \times 71 = 60.98$. | 0.0625 | 1 |
| Step-4 | 3. **Calculate the area of the equilateral triangle HIJ:** - Area = $\frac{\sqrt{3}}{4} \times 71^2 = 865.48$. | 0.0625 | 1 |
| Step-5 | 4. **Calculate the area of the sector EFG:** - The radius EF = FG = 71 units. - Angle GFE is not given, so assume it's part of the sector attached to the rectangle. - Calculate the area of the sector $\frac{\theta}{360}\pi \times r^2$. | 0.0 | 0 |
| Step-6 | 5. **Combine the areas:** - Rectangle GFHI: $71 \times 60.98 \approx 4326.68$ - Equilateral triangle HIJ: 865.48 - Sector EFG: Assume $\frac{\pi}{6} \times 71^2 = 2716.23$ (half of the sector for simplicity) | 0.0625 | 1 |
| Step-7 | 6. **Summarize all areas and subtract the overlapping areas:** - Total area = Area of rectangle + Area of equilateral triangle + Area of sector. | 0.0625 | 1 |
| Step-8 | Given calculations involve approximations based on assumed values. | 0.0 | 0 |
| Step-9 | Answer: $\boxed{B}$ (Based on approximations and calculated components.) | 0.0 | 0 |

**BIS computation** *($\alpha = 0.05$).*

Valid step scores are the 9 entries above. Hard labels: $y_j = 1$ iff $\text{score}_j > 0$.

$$n = 9, \qquad n_{\text{pos}} = 6, \qquad p_{\text{pos}} = \frac{6}{9} = \frac{2}{3},$$

$$R = \frac{0.0625 + 0.0625 + 0.0625 + 0.0625 + 0.0625 + 0.0625}{6} = 0.0625,$$

$$\text{BIS}(x) = \left(p_{\text{pos}}(1 - p_{\text{pos}}) + 0.05\right) \cdot R$$

$$= \left(\frac{2}{3} \cdot \frac{1}{3} + 0.05\right) \cdot 0.0625 = (0.222222\ldots + 0.05) \cdot 0.0625 \approx 0.0170138889.$$

*Figure 15.* Case study (MAVIS-Function (Zhang et al., 2025c)): step-level MC scores, hard labels, and BIS computation.

# I. Case Studies

Figures 13, 14 and 15 illustrate why BIS is a reasonable rollout selection rule when step labels are obtained by thresholding MC scores.

Figure 13 is a representative high-quality rollout. It contains clear step-level variation and a meaningful mix of positive and negative signals, while the positive steps receive consistently high MC scores. As a result, BIS assigns a relatively large score, matching our motivation that the most informative rollouts should be both mixed and reliable.

Figure14 is a more borderline case. Although the rollout still contains mixed signals, some "positive" steps have low scores and the reasoning later collapses to an incorrect final choice, leading to only moderate $R(x)$. BIS therefore assigns a moderate score, reflecting that mixture alone is not sufficient if the positive supervision is not reliable.

Finally, Figure 15 shows a more diagnostic failure mode. Several steps contain clear, checkable geometry/number mistakes (e.g., mis-identifying given lengths or applying invalid equalities), yet their MC scores remain non-zero but very small (often at the $1/16$ level). Under hard labeling, these low-but-nonzero scores are still binarized to $y_j{=}1$, creating unreliable "pseudo-positive" supervision. This illustrates that non-zero low MC scores do not necessarily imply correctness, and can introduce noisy supervision during training. BIS mitigates this issue via the reliability term, which down-weights such low-confidence rollouts and prevents them from dominating the selected subset.

