# OpenReview forum: "Training Data Efficiency in Multimodal Process Reward Models"
_ICML.cc/2026/Conference — ICML 2026 regular_

### Official Review · Reviewer_CpCa · 2026-03-01

**Soundness:** 2
**Presentation:** 3
**Significance:** 3
**Originality:** 3
**Overall Recommendation:** 4
**Confidence:** 3

**Summary:**

This paper introduces a data filtering framework for multimodal reward models. The authors empirically find out that the data redundancy issue in existing MC-annotated corpora. To address this, the authors propose a data filtering framework that selects the data based on label mixtures of positive/negative steps and label reliability. A theoretical framework is proposed to explain the rational behind the design choice. The experimental results on VisualProcessBench demonstrated that the proposed methods are effective compared with baselines.

**Compliance With Llm Reviewing Policy:**

Affirmed.

**Final Justification:**

The authors' rebuttal addresses concerns and I will keep my positive rating.

**Key Questions For Authors:**

- Lack of conceptual novelty. The core heuristic of selecting samples based on "mixture" and "reliability" is a well-established principle in classical statistical learning and active learning, e.g., Hard Negative Mining, Uncertainty Sampling. While the paper successfully adapts these concepts to the specific domain of multimodal reward models, the underlying theoretical contribution feels more like an application of existing learning theory rather than a fundamental algorithmic breakthrough.
- The experiment is limited to single training dataset. The authors conduct analysis and experiments only with VisualPRM400K-v1.1, which raise a concern that the observed phenomenon is a dataset specific problem and the proposed framework only works on this dataset.
- Concerns about the training stop critieria. In Table 2, the comparison between the proposed BIS method and baselines relies on a fixed, manually selected checkpoint. This raises several concerns regarding the fairness of the evaluation. For example, how the stop criteria is chosen? Is the test set directly used for this?

**Limitations:**

The limitation is not discussed.

**Strengths And Weaknesses:**

- This paper explores an important and practical problem, the data efficiency for training multimodal reward model.
- This paper is well-written and easy to follow.
- The proposed data filtering framework is effective under proposed experiments setting.

---

> ### Author Rebuttal · Authors · 2026-03-31
>
> **Q1. Novelty relative to prior selection principles**
>
> **A:** Thank you for this thoughtful comment. We agree that high-level concepts are related with classical statistical or active learning methods. However, they typically rely on extra online model evaluations to calculate uncertainty during training, and are not specifically designed for MC-annotations.
>
> Our contribution is not to claim a fundamentally new theoretical paradigm, but to explore empirical phenomena unnoticed by the research community in process reward modeling, and how to apply statistical principles (mixture and reliability) without extra compute cost. Specifically, we demonstrate that informative training data in MC-annotated corpora are not adequately characterized by classical heuristics such as prioritizing “hard” or “uncertain” samples. This is because MC annotation introduces unavoidable noise and instability, so effective selection must account for their interaction.
>
> Our analytical framework isolates this previously underexplored structure in process reward modeling, clarifies why this interaction between noise, the model, and the data must be explicitly modeled, and uses it to derive a simple rule tailored to the practical constraints of this regime. This leads to BIS that incurs essentially **no additional selection cost**, a particularly attractive property in practice.
>
> **Q2. Dataset-level generalization**
>
> **A:** Thank you for this important point. We agree that validation on additional MC-annotated corpora would further strengthen the paper. At the same time, we would like to clarify that VisualPRM400K-v1.1 is actually a collection of 38 distinct source datasets. Table 1 reports the overall scale of VisualPRM400K-v1.1 (**565,096** rollouts / **3,174,394** annotated steps), and Appendix Table 8 further breaks it down into 38 source subsets spanning diverse task families, including document/chart/diagram/VQA sources (e.g., AI2D, ChartQA, DocVQA, DVQA, FigureQA, IconQA, InfographicVQA, SROIE, VQAv2), synthetic/compositional reasoning sources (e.g., CLEVR, Super-CLEVR, NLVR2), and math/geometry/science/multimodal CoT sources (e.g., Geo170K, Geometry3K, GeoQA+, GEOS, UniGeo, ScienceQA, MathV360K, M3CoT, and the MAVIS family). Our experiments on this corpus validates BIS on a heterogeneous multimodal PRM training pool, rather than on a narrow domain.
>
> Moreover, our subset construction is source-stratified (preserving the source proportions): for each keep ratio, Random-ρ retains the same fraction from each of the 38 sources, and BIS applies selection within the same source-wise budget. Therefore, the gain is not driven by shifting toward a few favorable subsets, but by better rollout selection within a large heterogeneous corpus. We will clarify this in the revision and discuss broader cross-corpus validation as future work.
>
> **Q3. Stopping criteria and checkpoint selection**
>
> **A:** Thank you for raising this concern. **All methods** in Table 2 use the same training protocol (single-pass training with the budget scaled proportionally to the retained rollout pool) and the same checkpoint selection rule: we report the best checkpoint within each run under the corresponding budget. Thus, Table 2 is intended to compare the best attainable performance under a fixed budget for each data-selection strategy, rather than to simulate a deployment setting with a separate validation set for early stopping. Under this goal, the protocol is applied equally to BIS, Random, Full-Data, and the other baselines, and does not favor BIS specifically. This is also consistent with prior MPRM practice[1], which has no standard validation split.
>
> We also agree that relying only on the best checkpoint could be concerning if one method were unstable. Importantly, this is not the case here: the paper already reports the full training dynamics (Figs. 6,7,8,9 / Appendix G), where BIS consistently outperforms the baselines throughout training rather than only at a single checkpoint. Due to the space limit, we show one representative example (Fig.6) here and refer to the Figures 6-9 for details. In InternVL2.5-8B, fixed budget setting, the overall F1 across six checkpoints is:
>
> | Step |  Avg. |
> |---|---:|
> | Random-25% | 62.49 |
> | Low-MC-25% | 63.13 |
> | Mixed-25% | 64.32 |
> | Reliable-25% | 62.05 |
> | BIS-25% | **65.21** |
>
> Due to space, details are in Supplementary Table 5: https://anonymous.4open.science/r/Balance-Info-F83C
>
> BIS is the best at all checkpoints, and has the highest checkpoint-averaged performance. Therefore, the gain is not driven by a single checkpoint, but reflects a stable advantage throughout training.
>
> The above constitutes our complete response, which we hope resolves all uncertainties. Your expert feedback has been invaluable, and we deeply appreciate the time you have devoted to reviewing our manuscript.
>
> [1] Wang et al., VisualPRM400K

---

> > ### Author Rebuttal · Reviewer_CpCa · 2026-04-03
> >
> > Thanks the authors for the reply. My concerns are addressed and I will keep my positive rating.

---

> > > ### Author Response · Authors · 2026-04-07
> > >
> > > We are delighted that our response successfully addressed your concerns. We sincerely appreciate your insightful comments and the AC’s effort in coordinating the review process.
> > >
> > > Best regards,
> > >
> > > The Authors

---

### Official Review · Reviewer_1jo3 · 2026-03-10

**Soundness:** 2
**Presentation:** 3
**Significance:** 3
**Originality:** 2
**Overall Recommendation:** 4
**Confidence:** 4

**Summary:**

This paper studies data efficiency in training multimodal process reward models (MPRMs) from Monte Carlo (MC)-annotated rollouts. The authors first show empirically that random subsampling of VisualPRM400K-v1.1 results in rapid saturation, suggesting redundancy in existing MC-annotated datasets. They then propose a rollout-selection heuristic, the Balanced-Information Score (BIS), which combines rollout-level label mixture and a reliability term based on the average MC score of positive steps. Empirically, BIS is evaluated on two backbones (InternVL2.5-8B and Qwen2.5-VL-7B) and consistently outperforms random subsampling at matched budgets, often approaching or matching full-data performance using much smaller subsets. The paper also presents a teacher-student-style analysis intended to justify why “mixed but reliable” rollouts should be informative

**Compliance With Llm Reviewing Policy:**

Affirmed.

**Final Justification:**

I have read the authors rebutals carefully and appreciate the aditional experiments and discussion we had . The new pseudo-positive retention rates (BIS filtering far more aggressively than alternatives), the low correlation between mixture and within-rollout heterogeneity (r=0.0071), the matched-group stepwise uncertainty comparisons, the multi-seed results confirming consistent gains, and the strict 7.67% budget experiment collectively strengthen the empirical case. The two additional baselines (SemDeDup-style and stepwise uncertainty ranking) also help contextualize BIS's effectiveness, though the baseline coverage remains somewhat narrow.

That said, my core concern about the gap between the theoretical narrative and empirical mechanism has been partially but not fully resolved. The theory remains a post-hoc abstraction that motivates BIS rather than tightly explaining it.

On balance, BIS is simple, zero-cost, and demonstrably effective across multiple backbones, benchmarks, and budgets. The practical contribution is clear, and the theoretical motivation, while not airtight, is reasonable and well-presented.

**Key Questions For Authors:**

-  Can you provide direct mechanism validation for the proposed explanation?  For example, can you measure pseudo-positive rate, per-step label fidelity, gradient variance, or gradient alignment for BIS-selected subsets versus Random/Mixed/Low-MC?

- Can you compare BIS to stronger and more theory-matched data-selection baselines? Maybe uncertainty-only selection, diversity-aware selection, loss/gradient-based selection, average stepwise $s_j(1-s_j)$, or a selector based directly on the Beta-Binomial pseudo-positive posterior.

- How robust are the reported gains across seeds? Please report multiple runs, confidence intervals, or error bars, especially for smaller differences such as BIS-25% vs. Mixed-25%.

- Can you tighten the theory by separating the latent variables more carefully? In particular, please clarify the distinction between true step correctness, one-shot future solve probability, and MC-positive probability under $Y=\mathbf{1}[K>0]$, and state more explicitly when rollout mixture is expected to be a good proxy for average stepwise informativeness.

-  Why is $R(x)=1$ for rollouts with no positive steps, and why is only positive-step reliability modeled? Since negatives may also be noisy under the MC-binarized labeling rule, I would like to understand the motivation for this design choice and whether alternatives were tested.

**Strengths And Weaknesses:**

One clear strengh of the p[aper os that MPRM training is expensive, and reducing rollout requirements without hurting performance would be useful to the community. BIS is also appealing as a method because it is simple, cheap, and post hoc: it uses only MC statistics already present in the dataset and requires no extra model calls or relabeling. The empirical results support the narrow practical claim that BIS is an effective subset-selection heuristic in this training setup. n Table 2, BIS substantially improves over random subsampling across budgets and both backbones; for InternVL2.5-8B, BIS-10% reaches 65.46 overall micro-F1 versus 62.86 for Random-10% and slightly exceeds the full-data score of 65.12, while for Qwen2.5-VL-7B the gains at very small budgets are especially large (e.g., 64.42 vs. 53.54 at 5%).

My main concern is soundness of the explanatory story. The theory is readable and sensible, but in my view it mostly provides a stylized justification for BIS rather than a tight theory of MPRM training. The most informative part is the standard logistic/Bernoulli observation that the local learning signal scales with $q(1-q)$, which already appears in Eq. (5). That intuition is valid, but it is also classical: it is the same quantity that appears as Bernoulli variance, Fisher information, Hessian curvature of cross-entropy, etc.  Also the generic SGD excess-risk decomposition in Eq. (4) is much less informative, because it is broad enough that it can rationalize many saturation stories after the fact.

A more serious theoretical issue is that the paper’s rollout-level proxy is not the same as the stepwise quantity it claims to care about. The paper argues that rollout mixture $\hat p_x(1-\hat p_x)$ estimates a teacher-level uncertainty term $\theta_x=\bar q_x(1-\bar q_x)$, and Appendix F.4 proves only an upper-bound style relation between the average stepwise signal $A(x)=\frac1n\sum_j q_j(1-q_j)$ and $\theta_x$. But the exact relation is $A(x)=\theta_x-\mathrm{Var}_j(q_j)$. This means a rollout can look highly “mixed” at the rollout level while containing very little stepwise uncertainty if it is heterogeneous across steps. So BIS should be presented as a heuristic proxy, not as a close estimator of the stepwise signal. Similarly, the true object in their own derivation is an average of products across steps, while BIS is effectively closer to a product of rollout-level averages; any covariance between uncertainty and reliability is dropped.

The symmetric-noise analysis in Eq. (6) is mathematically fine as a second-moment expression, but the interpretation is incomplete because noisy labels introduce not only variance but also directional bias. Overall, I found the theory helpful for organizing intuition, but not strong enough to establish that BIS works specifically for the causal reasons emphasized in the paper.

The comparisons among Random-25%, Low-MC-25%, and Mixed-25% are not conlusive, yet those subsets differ in several things at once, including error-step ratio, average MC score, and rollout composition. Mixed-25% is also not a clean intervention on mixture, since naturally mixed rollouts are only 7.67% of the pool and the remainder is filled randomly.

the experiments never directly measure the mechanism claimed by the theory: there is no direct evidence that BIS-selected subsets have lower pseudo-positive rates, better label fidelity, lower gradient variance, or better gradient alignment. So the paper convincingly shows that BIS is useful, but it does **not** convincingly show that it has identified the precise internal reason why BIS works.

There are also several missing experimental controls that matter for my evaluation. First, the strongest baselines are mostly internal heuristics rather than stronger data-selection baselines from the broader pruning/subset-selection literature. Second, the detailed component ablation is essentially concentrated at one operating point (25%), and the paper does not show comparable breakdowns across more budgets and both backbones.

---

> ### Author Rebuttal · Authors · 2026-03-31
>
> **Q1. A:** Thank you for the question. First, we clarify that exact label fidelity/pseudo-positive rates are **unmeasurable**, because strict ground truth for solution steps in standard MC datasets is unavailable.
>
> Second, regarding gradient metrics: Our analysis uses an "ideal teacher" as a reference to isolate the intrinsic informativeness of the data, answering what kind of data is generally the most useful. Under this perfect model, gradient are driven purely by the quality of the data itself. But this ideal model is unobtainable because absolute step-level correctness is unknown; we only have access to noisy MC estimates. In actual training, the gradients observed from a student model mix true data signal with the student's own prediction errors and dynamically changing weights. Therefore, the measurable empirical gradient of the student model does **not** correspond to the static, intrinsic data value captured by our analysis.
>
> **Q2. A:** Due to space limitations, refer to reviewer VzKC's W2. Thanks.
>
> **Q3. A:** We thank the reviewer for raising this issue. We run the 25% InternVL setting across 3 different random seeds: BIS achieves 65.38±0.13, while Mixed achieves 64.37±0.29, showing that the BIS gain over Mixed is consistent. We further added 10% breakdowns on both backbones. The same pattern holds: BIS is the best on both backbones. This indicates that the advantage is not specific to the single budget or to a single backbone. Full details: https://anonymous.4open.science/r/Balance-Info-F83C
>
> **Q4. A:** They are three linked levels in the MC pipeline. The first concerns whether the step itself is correct, the second concerns whether the continuation from that step can solve the problem, and the third is the observed binary training label. We uses this chain to explain why the reliability of the observed label depends on the continuation success probability of that step, and where the noise in the observed label comes from.
>
> As clarified in Sec. 3.3, under noise-free labels the empirical rollout mixture is an observable
> O(1/n)-biased estimator of the rollout-level uncertainty term (Eq.7–9), and the average stepwise informativeness is upper-bounded by that same rollout-level quantity (Eq.8). Hence, we do not use mixture to claim that every mixed rollout must have large stepwise information; rather, low mixture rules out large average stepwise informativeness, while higher mixture leaves more room for informative gradients. The O(1/n) bias is also moderate in our setting since rollouts average ∼5.6 steps (Table 1). We then further account for the practical case that MC noise is unavoidable. As Eq. 10 shows, the observed rollout mixture remains a reliable proxy only when this noise is small. BIS therefore adds the reliability term to favor low-noise rollouts, so that the mixture signal more faithfully reflects the underlying uncertainty.
>
> **Q5. A:** We model reliability only for positive steps due to the MC binarization rule (positive if $\ge1$ of 16 continuations succeeds). Thus, $P(Y=1\mid r)=1-(1-r)^{16}$ and $P(Y=0\mid r)=(1-r)^{16}$, where $r$ is the one-shot success rate. Even small $r$ can trigger a positive label: if $r=0.01$, $P(Y=1)\approx 0.15$. In contrast, a negative label requires all continuations to fail: $P(Y=0)\approx 1.5\times10^{-5}$ for $r=0.5$. Hence, low-MC pseudo-positives are the dominant noise source, motivating positive-step reliability only (Sec 3.3).
>
> For all-negative rollouts, positive-step reliability is undefined: there is no positive step to average over, and no extra reliability signal is available. we set a neutral $R(x)=1$; with $p_{\text{pos}}=0$, BIS reduces to $\alpha$, so these rollouts are not favored.
>
> **W. Comparisons among exploratory subsets**
>
> **A:** These are exploratory preliminary experiments designed to discover data patterns, not strict causal ablations; this exploration itself is a primary contribution of our work. Cleanly isolating every factor is neither necessary—as these findings already successfully identify the attributes of high-quality data—nor possible. In real datasets, properties like average MC scores and error-step ratios are coupled statistical features that cannot be independently manipulated.
>
> **W. Positioning and scope of the theoretical explanation**
>
> **A:** The goal of Sec. 3 is not to present BIS as a tight estimator of the exact stepwise signal. Rather, it adopts simplified but analyzable abstraction to extract the key properties of informative data and to turn them into a simple, practical criterion. A more exhaustive treatment of every detail would likely introduce additional complexity without serving this goal. More importantly, our contribution is not confined to Sec. 3: it also includes the preliminary identification of the previously underappreciated empirical phenomenon in MPRM training (Sec.2), and a simple method derived from this understanding whose effectiveness is validated across models and settings (Sec.4&5).

---

> > ### Author Rebuttal · Reviewer_1jo3 · 2026-04-02
> >
> > I have the following questions for the authors:
> >
> > -  **Regarding Q1**: I dont agree entirely that exact label fidelity/pseudo-positive rates are unmeasurable. For instance, the authors could measure the distribution of MC scores among positive-labeled steps in BIS-selected vs. random subsets (a direct proxy for pseudo-positive rates), or compute empirical gradient norms/variance during early training. Your claim that this does not correspomnd to static intrinsic data value, while tru feels like it sidesteps the issue, if the theory correctly identifies the mechanism, you'd still expect observable correlations between BIS selection and lower gradient noise in practice.
> >
> > - **Regarding Q4**: I feel here like the authors mostly restated what was already in the paper without engaging with my question. For instance, you acknowlegde that rollout mixture can be high while stepwise uncertainty is low (heterogeneous rollouts) but then you also say this is an upper-bound proxy and frame this is as sufficient because "low mixture rules out large informativeness.". This feel true but i argue its still weak.  it means BIS could be selecting many rollouts that look mixed but aren't actually informative at the step level. ON the exploratory subsets you used, you acknowledge these aren't clean causal ablations but argue they don't need to be. This is fair, since real datasets have coupled properties, but it weakens the paper's explanatory narrative. If the subsets can't cleanly isolate factors, the theory sections shouldn't present them as evidence for specific causal claims. On the position of the theory, the authors now frame Section 3 as providing "simplified abstraction" rather than tight theory, which is more honest, but the paper still presents the theory as explaining why BIS works. This kind of makes the contributions thinner.

---

> > > ### Author Response · Authors · 2026-04-07
> > >
> > > **Q1:**
> > >
> > > We appreciate the reviewer's follow-up. To clarify, our previous response aimed to address the original request for a direct label-fidelity measurement, which is unmeasurable because exact step-level ground truth inherently does not exist for 3.17M reasoning steps.
> > >
> > > We agree with the new follow-up suggestion that empirical results can serve as a meaningful proxy. Leveraging MC scores as noise indicators is a core insight formalized in paper Sec. *MC Scores as Effective Noise Indicators*. Fig 3,11,12 already demonstrate that BIS systematically suppresses low-MC steps across all 38 source datasets.
> > >
> > > To explicitly compare this across different selection strategies, we quantified the retention rate of the lowest mc score positive step (k=1,2/16) in the link. The rates are: BIS(**19.1%**), Random(25.4%), Mixed(71.8%), Low-MC(88.1%), confirming BIS's superior ability to filter pseudo-positives.
> > >
> > > Regarding the gradient part of Q1, empirical gradients are heavily entangled with initialization, optimizer/lr dynamics, batch composition, representation geometry, etc. They do not provide a clean measurement of model-independent properties studied here.
> > >
> > > However, training loss (Fig.2 in link) provides the ultimate validation: BIS consistently achieves the lowest loss, confirming the practical correlation between BIS selection and reduced gradient noise.
> > >
> > > **Q4a:**
> > >
> > > To address the concern that our upper-bound proxy (Mixture) might select uninformative "heterogeneous" rollouts, we demonstrate that M does not systematically bias selection toward heterogeneity, and empirical stepwise informativeness is further secured by the Reliability term.
> > >
> > > * Among mixed rollouts, the Pearson correlation between label mixture M=p_{pos}(1-p_{pos}) and within-rollout step-score variance Var_s=Var_j(s_j) is only **0.0071**. If high mixture were primarily driven by heterogeneous steps, M and Var_s would be strongly positively correlated. We do not observe such a trend. While heterogeneous cases exist, prioritizing M does not systematically bias the selection toward them.
> > >
> > > * To ensure the selected rollouts are genuinely informative at the step level, BIS incorporates R. Analyzing matched high-M rollouts (top 25% M), the high-M_high-R group exhibits significantly stronger empirical stepwise uncertainty proxies ($U_1 = \text{avg}[s_j(1-s_j)]$ and $U_2 = \text{avg}[\min(s_j, 1-s_j)]$) than the high-M_low-R group: $U_1$ (0.0735 vs. 0.0517) and $U_2$ (0.1080 vs. 0.0616). This shows that, within matched high-M candidates, higher R is associated with stronger empirical stepwise uncertainty proxies.
> > >
> > > In summary, ~3.17M corpus statistics confirm M does not systematically attract heterogeneous failure modes, and R further secures stepwise informativeness.
> > >
> > > **Q4b:**
> > >
> > > We are glad to reach a consensus: error-step ratio and average MC score are essentially two sides of the same coin: rollouts with more zero-MC error steps lower the average MC score. However, we took an extra step to strictly isolate the only factor we could, evaluating all subsets at a strict 7.67% budget in link (the exact amount of mixed rollouts). In this setting, BIS still performs the best. (**BIS-64.74**;Random-61.63;LowMC-62.25;Mixed-63.22)
> > >
> > > Furthermore, we clarify our logical flow: these subsets aren't "evidence for causal claims," nor do we frame them as such. They reflect our heuristic exploration in a noisy setting; sharing this process aims to reveal dataset-intrinsic property.
> > >
> > > As stated in Line133, our analysis aims to explain Empirical Findings 1&2. Sec.2 presents the phenomenon, while Sec.3 provides a math abstraction to interpret it. The validity of Sec.3 rests entirely on its internal derivations, not on the subsets from Sec.2. We hope this clarifies our narrative.
> > >
> > > **Q4c:**
> > >
> > > We respectfully disagree that a simplified abstraction diminishes contribution. As explicitly stated (e.g.,providing a "practical recipe with grounded analysis" Line79), our work is a data-centric empirical study. Sec.3 serves as a grounded theoretical lens, not a monolithic exact proof.
> > >
> > > Deep learning breakthroughs frequently stem from abstractions distilling complex realities. [1] used linear networks to reveal deep non-linear training dynamics. [2] leveraged tractable linear frameworks to explain CNN adversarial examples. Our methodology aligns strictly with this established practice.
> > >
> > > Our contribution rests on effectiveness. This abstraction motivated BIS, a strictly zero-cost method cutting 3M+ training data to 10% without performance loss. We validate this across 3 model families/scales, 8 benchmarks and 6 retention ratios. Solving this efficiency bottleneck represents a substantial, practical contribution.
> > >
> > > The above constitutes our complete response, which we hope resolves all remaining uncertainties. We appreciate the time and care you have devoted to reviewing our manuscript.
> > >
> > >
> > > [1] Saxe et al., ICLR, 2014.
> > >
> > > [2] Ilyas et al., NeurIPS, 2019.

---

### Official Review · Reviewer_VzKC · 2026-03-12

**Soundness:** 3
**Presentation:** 3
**Significance:** 3
**Originality:** 3
**Overall Recommendation:** 5
**Confidence:** 4

**Summary:**

This paper studies data efficiency for training Multimodal Process Reward Models (MPRMs). They develop a theoretical framework showing that informative gradient updates depend on two factors: label mixture (balance of positive/negative steps) and label reliability (MC scores of positive steps). Based on these insights they propose Balanced-Information Score (BIS), a rollout-level scoring criterion for data selection that prioritizes both mixture balance and reliability, using only existing MC signals without extra cost. Experiments on InternVL2.5-8B and Qwen2.5-VL-7B show BIS-selected subsets match full-data performance with only 10% of data.

**Compliance With Llm Reviewing Policy:**

Affirmed.

**Final Justification:**

The authors have addressed my conerns and provided detailed additional experimental results as requests. Thus, I'll increase my final score by 1

**Key Questions For Authors:**

1. How sensitive is BIS to the quality of the initial MC annotations? If the MC estimates themselves are noisy (which they often are with limited rollouts), does BIS degrade gracefully or does it amplify selection errors? Can you conduct agreement study on this?

2. Regarding the step-to-rollout aggregation: have you experimented with step-level selection (selecting individual steps rather than full rollouts)? The theory suggests step-level informativeness varies, so rollout-level selection might be suboptimal.

3. How does the 10% threshold change across different training set sizes? If you started with a smaller or larger corpus, would BIS still converge to ~10%?

**Limitations:**

Yes, the authors discuss limitations adequately.

**Strengths And Weaknesses:**

## Strengths
- Strong theoretical grounding. The paper derives the gradient variance decomposition and connects it to practical quantities (label mixture, MC reliability). The progression from Theoretical Finding 1 through 4 builds a coherent story. This is not just an empirical trick but has principled justification.

- Practical contribution. Reducing training data to 10% with no performance loss is genuinely useful given the cost of MC annotation. The BIS score is simple to compute and adds zero overhead.

- Clean experimental design. The saturation curves in the preliminary experiments clearly motivate the work. The comparison against random subsampling and other baselines across two different backbones demonstrates robustness.

- Well written paper overall, I would say with a good balance between theory and experiments.


## Weaknesses
- The symmetric label noise approximatio is a strong simplification. The authors acknowledge step-dependent noise but don't analyze how well the constant-noise theory matches practice. Some empirical validation of this assumption would strengthen the theoretical contribution.

- The paper focuses on the data selection aspect but doesnt compare against other data-efficient training strategies like curriculum learning, active learning, or coreset methods. The BIS approach is essentially a scoring function for static selection — how does it compare to adaptive/online methods?

- Limited scale. Both backbones are ~7-8B parameter models. Its an open question whether the same 10% efficiency holds for larger models like 14 or 32B models, and the theoretical framework doesnt directly address this.

- The connection between the rollout-level BIS score and the step-level theoretical analysis could be made tighter. There seems to be a gap where the theory operates at the step level but the practical method selects entire rollouts.

---

> ### Author Rebuttal · Authors · 2026-03-31
>
> **Q1: MC-noise Setting**
>
> **A:** We appreciate the attention to the central challenge of our setting. The inherent noise in MC annotations is exactly the core motivation behind our analysis and the design of BIS. We do not assume clean MC estimates. As analyzed in Sec. *MC Scores as Effective Noise Indicators*, mixture alone cannot separate informative samples from low-confidence pseudo-positives under noisy MC estimation. So BIS adds an MC-based reliability term to suppress such noise. Empirically, Table 4 shows that BIS outperforms single-metric heuristics—both selecting harder but less reliable positives and selecting solely by reliability.
>
> A rigorous sensitivity study would require large-scale re-annotation under different noise settings (e.g., weaker estimators). Since the original dataset cost over 2000 A100 GPU hours for MC annotation [1], this is infeasible during rebuttal. We will add this to the limitation section.
>
> **Q2&W4: About Rollout Aggregation**
>
>  **A:** Thank you for this comment. We clarify that this is a deliberate design choice. Our analysis starts at the step level because the step is the smallest unit of labeling and supervision. The paper then does not jump directly to rollout-level selection, but introduces rollout-level proxies: label mixture reflects aggregated step-level information, and the average MC score over positive steps reflects reliability. BIS selects rollouts based on these two quantities.
>
> We perform selection at the rollout level because it is more compute-efficient in practice. Each retained rollout provides supervision for multiple steps, preserving denser supervision. By contrast, step-level selection may pick only a few scattered steps from many rollouts, making supervision sparser; under the same training budget, it may also require covering many more rollouts/problems, which means more forward passes and less compute efficiency. Besides, rollout-level aggregation remains consistent with the training pipelines used in prior PRM work [1], which provide full rollouts and dense training signals.
>
> **Q3&W3: 10% data efficiency across corpus and model scales**
>
>  **A:** We agree that 10% is not a universal threshold: BIS reaches full-data performance at a small keep ratio, but the exact ratio depends on the model (e.g., 10% for InternVL2.5-8B vs. 25% for Qwen2.5-VL-7B in Table 2) and also on corpus redundancy.
>
> We additionally ran a new InternVL3-14B experiment, and BIS-10% (66.05 overall-f1) still matches full-data (65.80) on larger backbone. This run required over 500 A100 GPU hours, but we felt it was important to directly address the reviewer’s concern for larger-scale validation. Due to space, details are in Table 1: https://anonymous.4open.science/r/Balance-Info-F83C
>
> As for corpus size, to the best of our knowledge, no other public (M)PRM dataset provides MC step scores, so a clean corpus-scale study is infeasible in rebuttal. Still, since VisualPRM400K has 38 source subsets and our selection is source-stratified, the current setup already gives partial evidence that BIS works beyond only the pooled corpus.
>
> **W1. Noise assumption in practice**
>
>  **A:** We appreciate the reviewer highlighting the issue of step-dependent noise. The constant noise $\eta$ in Sec. 3.3 serves as a mathematical stepping stone to derive the closed-form gradient variance (Eq. 6). As detailed in Theoretical Finding 5, we immediately relax this assumption via the Beta-Binomial model, transitioning to a dynamic effective noise $\eta_{eff}(k)$. Our final BIS relies entirely on this step-dependent noise. We agree that empirical validation is important. Table 4 supports this dynamic analysis: the performance drop on the Low-MC subset validates our noise analysis.
>
> **W2. Broader data-efficiency baselines**
>
>  **A:** We added two baselines under the same fixed budget:
>
> 1. a more general diversity-aware selection baseline, based on the classic framework [2], and conceptually related to coreset-style methods. We extract rollout embeddings using InternVL8B, perform semantic deduplication, and then keep representative samples (see Algorithm 1 at the link).
>
> 2. a problem-specific selector, which follows reviewer 1jo3’s suggestion and ranks rollouts by the average stepwise uncertainty $\frac{1}{n}\sum_j s_j(1-s_j)$.
>
> The resulting overall scores are 64.32 and 64.13, respectively, both below 65.46 of BIS. (details in Tables 2 at the link) Importantly, SemDeDup-style diversity-aware methods, as well as loss/gradient-based or adaptive/online methods, typically require substantial extra computation. BIS incurs essentially zero extra cost, and relies on the observable selection signal. Such a combination of low cost and effectiveness is uncommon in data selection.
>
> These additional results were obtained with substantial extra resources allocated on short notice. We are grateful that our work is understood. Thanks again!
>
> [1] Wang et al., VisualPRM400K, ICLR 26
> [2] Tirumala et al., D4, NeurIPS 23

---

> > ### Author Rebuttal · Reviewer_VzKC · 2026-04-02
> >
> > The author has actually addressed most of my concerns by extending the baseline models, explaining more details, and referring to previous papers. One small expectation is to see more detailed additional experimental results instead of single numbers.

---

> > > ### Author Response · Authors · 2026-04-02
> > >
> > > We are glad that the reviewer’s concerns have now been resolved. We sincerely value objective and evidence-based feedback, because it is exactly this kind of feedback that helps researchers better communicate previously unrecognized phenomena and insights to the research community.
> > >
> > > Due to the 5,000-character ICML rebuttal limit (i.e., roughly under 1,000 words), in our first  response, following the official ICML guidance, we attached the anonymous link and cited it in the first response for the full added tables and figures, while keeping only the most decision-relevant summary numbers inline for readability. Since we now have a second response window with a new 5,000-character allowance, we provide the detailed added results directly below, so that the evidence can be inspected here without requiring you to consult the anonymous link.
> > >
> > > - **Larger-scale validation on the 14B model.**
> > >
> > > To directly address the reviewer’s question on larger-scale validation, we additionally conducted an experiment on InternVL3-14B under the same evaluation protocol. We used InternVL3 rather than InternVL2.5 because InternVL2.5 does not provide a 14B-scale model.
> > >
> > > *VisualProcessBench results on InternVL3-14B. The **10% subset selected by BIS** achieves performance comparable to **full-data training**.*
> > >
> > > | Method | Overall | MathVision | MathVerse | MMMU | DynaMath | WeMath |
> > > |---|---:|---:|---:|---:|---:|---:|
> > > | BIS-10%  (InternVL3-14B)  | **66.05** | **67.39** | **66.34** | 62.65 | **66.41** | 64.21 |
> > > | Full-Data   (InternVL3-14B) | 65.80 | 66.87 | 66.23 | **63.75** | 65.37 | **64.31** |
> > >
> > > Thus, beyond the two backbones already evaluated in the main paper (InternVL2.5-8B and Qwen2.5-VL-7B), the effectiveness of our method is further validated on InternVL3-14B.
> > >
> > > - **Additional Baselines.**
> > >
> > > To directly address the reviewer’s question on broader data-efficiency baselines, we additionally implemented two representative baselines under the same 25% training budget on InternVL2.5-8B and report their full results below.
> > >
> > > *SemDeDup baseline (Simplified Explanation):* We extract rollout embeddings, then apply PCA whitening and $\ell_2$ normalization. We partition rollouts by source bucket (i.e., the 38 source subsets in VisualPRM400K, each treated as one bucket) and assign the same keep ratio to each bucket. We then remove near-duplicates within each bucket using a similarity threshold, resolve highly similar cross-bucket pairs while preserving bucket quotas, and finally fill the remaining quota by greedily selecting the least similar survivors.
> > >
> > > More detailed implementation is provided in Algorithm 1 of the anonymous link: https://anonymous.4open.science/r/Balance-Info-F83C. This baseline is adapted from D4: Improving LLM Pretraining via Document De-Duplication and Diversification (NeurIPS 2023).
> > >
> > > *Evaluation on VisualProcessBench for InternVL2.5-8B with two additional baselines.*
> > >
> > > | Method | Overall | MathVision | MathVerse | MMMU | DynaMath | WeMath |
> > > |---|---:|---:|---:|---:|---:|---:|
> > > | Average Stepwise-25% | 64.32 | 64.85 | 64.29 | 60.91 | 65.73 | 63.82 |
> > > | SemDeDup-25%         | 64.13 | 65.25 | 63.82 | 60.01 | 65.28 | 64.33 |
> > > | BIS-25%              | **65.46** | **67.98** | 64.86 | 60.49 | 65.72 | **65.59** |
> > > | Full-Data            | 65.12 | 65.77 | **65.43** | **61.84** | **66.17** | 63.56 |
> > >
> > > Classical data-selection algorithms such as SemDeDup typically require substantial additional computation, yet do not yield meaningful gains. By contrast, BIS incurs near-zero extra cost while achieving stronger performance. This experiment further highlights the uniqueness of our method: valuable subsets cannot be easily identified by single metric alone, and a classical data-selection algorithm also becomes less effective in the setting where labels are obtained via Monte Carlo estimation. This suggests, first, that our method captures structure genuinely related to informativeness; and second, that it is significantly effective under realistic training-budget constraints.
> > >
> > > For completeness, the anonymous link also includes several further results not reproduced here due to space: multi-seed repeated runs that verify robustness, additional ablations across multiple keep ratios and model families that further validate effectiveness, and checkpoint-wise results showing that BIS maintains its advantage throughout training.  https://anonymous.4open.science/r/Balance-Info-F83C
> > >
> > > We sincerely hope that the detailed tables above, together with these supplementary results, make the evidence more complete and easier to inspect. If you find the current evidence sufficient, we would be most grateful if you would take it into account in your final recommendation. We also sincerely thank Reviewer VzKC again for the valuable time and effort devoted to reviewing our work.

---

### Decision · Program_Chairs · 2026-04-30

**Decision:**

Accept (regular)

**Comment:**

This paper addresses a critical bottleneck in training Multimodal Process Reward Models, the high cost of Monte Carlo annotated data. The authors identify significant redundancy in existing datasets and propose the Balanced-Information Score (BIS), a zero-cost method that selects the most informative rollouts by balancing label mixture and reliability.
The work is highly impactful because it demonstrates that BIS can match or even exceed full-data performance using only 10% of the training data. This efficiency holds across multiple model scales, including up to 14B parameters, and diverse benchmarks, making it a very practical tool for the community.

During the discussion phase, the authors went above and beyond to address reviewer concerns. They provided strong empirical evidence that their rollout-level method effectively captures step-level informativeness and included impressive new baselines, such as diversity-aware selection, which BIS outperformed despite having zero additional computational overhead.
While some reviewers questioned the tightness of the theoretical framework, the authors bridged this gap through detailed correlation studies and proxy measurements of pseudo-positive rates.

The consensus among the reviewers is that the paper is technically sound and offers a practical contribution to efficient multimodal reasoning. The authors have successfully addressed all major technical concerns through extensive additional experiments.